# Identifying Axiomatic Mathematical Transformation Steps using Tree-Structured Pointer Networks

**Sebastian Wankerl**                                    *wankerl@informatik.uni-wuerzburg.de*
*Center for Artificial Intelligence and Data Science (CAIDAS)*
*Julius-Maximilians-Universität Würzburg, Germany*

*Baden-Württemberg Cooperative State University Mosbach, Germany*

**Jan Pfister**                                          *pfister@informatik.uni-wuerzburg.de*
*Center for Artificial Intelligence and Data Science (CAIDAS)*
*Julius-Maximilians-Universität Würzburg, Germany*

**Andrzej Dulny**                                        *andrzej.dulny@uni-wuerzburg.de*
*Center for Artificial Intelligence and Data Science (CAIDAS)*
*Julius-Maximilians-Universität Würzburg, Germany*

**Gerhard Götz**                                         *gerhard.goetz@mosbach.dhbw.de*
*Baden-Württemberg Cooperative State University Mosbach, Germany*

**Andreas Hotho**                                        *hotho@informatik.uni-wuerzburg.de*
*Center for Artificial Intelligence and Data Science (CAIDAS)*
*Julius-Maximilians-Universität Würzburg, Germany*

**Reviewed on OpenReview:** *https://openreview.net/forum?id=gLQ8O1ewwp*

## Abstract

The classification of mathematical relations has become a new area of research in deep learning. A major focus lies on determining mathematical equivalence. While previous work has simply approached the task as a binary classification without providing further insight into the underlying decision, we aim to iteratively find a sequence of necessary steps to transform a mathematical expression into an arbitrary equivalent form. Each step in this sequence is specified by an axiom together with its position of application. We denote this task as Stepwise Equation Transformation Identification (SETI) task. To solve the task efficiently, we further propose *TreePointerNet*, a novel architecture which exploits the inherent tree structure of mathematical equations and consists of three key building blocks: (i) a transformer model tailored to work on hierarchically tree-structured equations, making use of (ii) a copy-pointer mechanism to extract the exact location of a transformation in the tree and finally (iii) custom embeddings that map distinguishable occurrences of the same token type to a common embedding. In addition, we introduce new datasets of equations for the SETI task. We benchmark our model against various baselines and perform an ablation study to quantify the influence of our custom embeddings and the copy-pointer component. Furthermore, we test the robustness of our model on data of unseen complexity. Our results clearly show that incorporating the hierarchical structure, embeddings and copy-pointer into a single model is highly beneficial for solving the SETI task.

# 1 Introduction

Deep learning has surpassed traditional machine-learning methods in multiple domains such as computer vision, natural-language processing, speech recognition and many more (Wang et al., 2020; González-Carvajal & Garrido-Merchán, 2021; O'Mahony et al., 2020). Also the field of symbolic mathematics has seen deep learning models applied to various objectives (Lu et al., 2023), such as recognizing equivalent mathematical expressions (Wankerl et al., 2021; Arabshahi et al., 2019; Mali et al., 2021; Wankerl et al., 2023) or performing symbolic computations for, e.g., integration (Lample & Charton, 2019), linear algebra (Charton, 2022), and solving recurrent equations (D'Ascoli et al., 2022). In addition, neural networks proved to be useful for generating step-by-step solutions to word problems (i.e. mathematical problems where the problem is partly formulated using natural language) and mathematical proofs (Azerbayev et al., 2024; Yu et al., 2024; Hendrycks et al., 2021).

The above-mentioned tasks of recognizing two equivalent mathematical expressions and of generating step-by-step solutions provide an inspiration for our work. Precisely, we want to find a sequence of verifiable axiomatic steps that is needed to transform a mathematical expression into another given equivalent form. We denote this task as Stepwise Equation Transformation Identification (SETI) task.

## 1.1 Introduction to the SETI Task

As an introductory example to the SETI task, consider the following pair of equivalent expressions: $x \cdot (y \cdot x)$ and $x^2 y$. For humans, it is easy to see that the right expression can be derived from the left by first applying the commutative law on the multiplication in the parentheses, yielding $x \cdot (x \cdot y)$, then applying the associative law, yielding $(x \cdot x) \cdot y$, and finally rewriting $x \cdot x$ as $x^2$, yielding $x^2 \cdot y$.

A sequence of such steps describes the transformation from one expression into the other. We observe that the SETI task can be solved iteratively, by only predicting the next step towards transforming one equation into another. This method (be it a human, a rule-based system or a neural network) is then applied on the input equation and its intermediate predictions until the target equation is reached. Thus, for a given pair of mathematical expressions we want to predict which axiom applied at which position of the expression tree yields a suitable step towards transforming the left-hand equation into the right-hand one.

Figure 1 provides a visualization of this setting. Note that the nodes of the left subtree (orange) are enumerated by a consecutive index. However, its only purpose is to make the nodes distinguishable for any method to extract the position at which the axiom was applied. It does not denote a different meaning regarding the mathematical interpretation, i.e. both $x_0$ and $x_1$ signify the same variable $x$, but at a different position.

Given a large enough equation and set of axioms, the number of applicable axioms at each step rapidly increases. Empirically, deep learning proved to be beneficial for a wide range of problems with large search spaces (Silver et al., 2016; Vinyals et al., 2015). Hence, we think that neural networks can be a useful heuristic to select a set of reasonable transformation steps. However, there are multiple hurdles when applying existing models to our task.

## 1.2 Limitations of Previous Research

So far, previous research has treated equivalence as a mere classification task without generating intermediate steps (Wankerl et al., 2021; Arabshahi et al., 2019; Mali et al., 2021; Wankerl et al., 2023). However, the classification task does not yield an explanation for the equivalence and depending on the complexity of the expression and the knowledge of the user, it might not be easy to see. Moreover, in remotely related settings like deep-learning-based step-by-step solutions to word problems, only the final answer is checked, but not the predicted intermediate steps (Azerbayev et al., 2024; Yu et al., 2024).

In this work, we want to provide an approach for overcoming these limitations. Besides, the latter task is generally approached using large language models (LLMs) which are trained for multiple purposes and consequently consist of billions of trainable parameters, making them computationally demanding for both fine-tuning and inference.

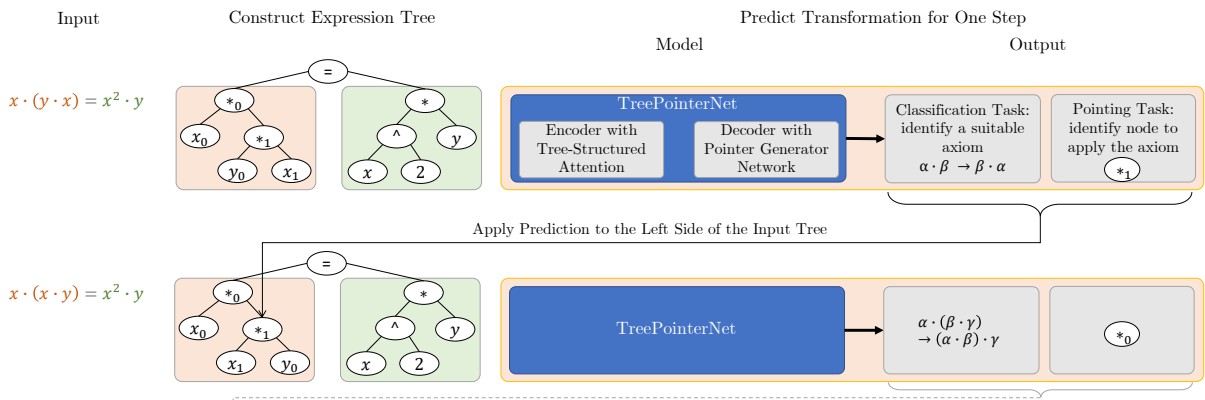

Figure 1: Overview of our SETI task: Given an input equation, the left side should be transformed into the right side by repeated application of TreePointerNet. At each step, the net predicts a suitable axiom and the root node of the subtree to apply the axiom. Note that for disambiguation of the mathematical tokens every node in the left subtree is combined with its incremental count to make it unique.

### 1.3 Contributions

To solve the SETI task more efficiently than LLMs, we introduce a new neural network architecture, *Tree-PointerNet*, designed specifically to solve both parts of our SETI task. It combines a tree-structured transformer encoder (Nguyen et al., 2020) with a pointer augmented decoder (Vinyals et al., 2015) and a custom many-to-one embedding layer that captures the semantic of the tokens independent of the position while making them still distinguishable for the pointer decoder.

Furthermore, we create and release a new dataset, consisting of equations and axiomatic steps needed to show their equivalence. This is necessary since to the best of our knowledge no well-suited dataset for the SETI task exists in literature. Our dataset contains a wide range of mathematical functions, such as polynomials, logarithms, exponentiation, and trigonometric functions. In addition, it contains equations of varying complexity measured by the depth of the parse tree and the number of axiomatic steps needed to show the equivalence.

Our contributions can be summarized as follows: (i) We introduce the SETI task in deep learning on symbolic mathematics aiming at explaining the relation of two equivalent mathematical expressions using axiomatic steps (ii) We design a novel neural network architecture to solve the proposed task efficiently and compare it to various neural networks known from literature (iii) We create a well-suited dataset for this task with varying complexity. The code for our experiments is available at `https://github.com/LSX-UniWue/axiomatic-steps-treepointer`.

## 2 TreePointerNet Architecture

Our goal is to identify mathematical transformations as depicted in fig. 1 and described in section 1.1. To inherently and efficiently capture the structure of mathematical expressions, we base our TreePointerNet on the tree-structured transformer model by Nguyen et al. (2020) (further denoted as TreeTransformer). It consists of an encoder which in contrast to a standard transformer (Vaswani et al., 2017) working on flat, i.e. sequential input, allows to input a tree. Thus, we can explicitly encode the hierarchy of mathematical operators and the link between operators and operands. Moreover, it is also equipped with a decoder that we can use to predict the applied axiom.

To identify the particular root node of the subtree that should be transformed, we propose a tree-copy-pointer mechanism. Hence, the model can be trained to directly extract any arbitrary node occurring in the input tree. A model without a copy-pointer component would be restricted to predicting nodes observed during training. We merge the tree-copy-pointer distribution with the decoder output of the TreeTransformer using

a gating layer. It learns to switch between copying nodes from the input tree and predicting axioms at each decoding step.

The first step of the model is to project the input tokens to an embedding. Generally, an individual embedding is learned for every distinct token. However, we represent each occurrence of a mathematical token as a token on its own. Consequently, this would lead to different and independently learned embeddings, although they actually represent the same mathematical entity. Thus, we incorporate a custom embedding layer which allows identical mathematical tokens to share one embedding over all their occurrences while still being unambiguously identifiable.

In the rest of this section, we describe our TreePointerNet architecture in detail as we designed it for our task. A visualization of the model is given in appendix A.1.

## 2.1 Background: TreeTransformer

Nguyen et al. (2020) propose an extension of the transformer that is incorporating tree-structured input into the attention mechanism. Precisely, their architecture receives a tree as input (encoder) and outputs a sequence of tokens (decoder). In this section, we give a short overview of the architecture. For more details, we refer to the original work.

Given a tree $\mathcal{T}$ specified as a set of $a$ leaves $\mathfrak{L} = \{x_0^{\mathcal{L}} \ldots x_a^{\mathcal{L}}\}$, $b$ non-terminal nodes $\mathfrak{N} = \{x_0^{\mathcal{N}} \ldots x_b^{\mathcal{N}}\}$ and a relation $\mathcal{R}(x)$, mapping each node $x$ to the set of all nodes that are successors of $x$ in $\mathcal{T}$, including $x$ itself.

The tree is first decomposed into two tensors corresponding to the embeddings of its $a$ ordered leaves $[l_1 \ldots l_a] = \mathcal{L} \in \mathbb{R}^{a \times d}$, and $b$ non-terminal nodes $[n_1 \ldots n_b] = \mathcal{N} \in \mathbb{R}^{b \times d}$, where $d$ denotes the size of the embedding.

Following this, a new tensor $S \in \mathbb{R}^{(b+1) \times a \times d}$ is constructed from $\mathcal{L}$ and $\mathcal{N}$ as

$$S_{i,j} = \mathcal{F}(\mathcal{L}, \mathcal{N}, \mathcal{R})_{i,j} = \begin{cases} l_j & \text{if } i = 1 \\ n_{i-1} & \text{else if } x_j^{\mathcal{L}} \in \mathcal{R}(x_{i-1}^{\mathcal{N}}) \\ 0 & \text{otherwise.} \end{cases} \tag{1}$$

Each column $S_{\cdot j}$ contains the hidden representations for all nodes on the path from the root of the tree to the $j$-th leaf $x_j^{\mathcal{L}}$. Row $S_i$ contains representations of the $i$-th node $x_i^{\mathcal{N}}$ in every column $j$ if it is on the path from the root node to the $j$-th leaf $x_j^{\mathcal{L}}$, otherwise zero.

As a next step, the bottom-up cumulative average $\hat{S}_{ij} = [\mathcal{U}(S)]_{ij} = \frac{1}{i}(S_{0,j} + \cdots + S_{i,j})$ is calculated, assigning to $S_{ij}$ the average over all values below itself. Finally, branch-level embeddings $\bar{n}_i$ representing each subtree rooted in non-terminal node $x_i^{\mathcal{N}}$ are obtained, resulting in tensor $\bar{\mathcal{N}} = [\bar{n}_1, \ldots, \bar{n}_b] \in \mathbb{R}^{b \times d}$. Each $\bar{n}_i$ is calculated as the weighted average of the $i$-th row over all non-zero columns using $\mathcal{V}$ as

$$\bar{n}_i = \mathcal{V}(\hat{S}, w)_i = \frac{\sum_{j=0}^{a} w_j \odot \hat{S}_{i,j}}{\mathcal{L} \cap \mathcal{R}(x_i^{\mathcal{N}})}. \tag{2}$$

Above described process is called *hierarchical accumulation*. Furthermore, since $\bar{\mathcal{N}}$ is a tensor consisting of all possible subtree embeddings, it is able to capture the global tree structure but not the local neighborhood of nodes and their relative positioning. To alleviate this limitation, hierarchical positional embeddings $E$ are added to each non-terminal node $x_i^{\mathcal{N}}$. They can be interpreted as a generalization of the positional embeddings on the sequential input of a transformer. They capture the absolute position of each non-terminal node $x_i^{\mathcal{N}}$ through a concatenation of two separate embeddings: The first embedding captures the number of nodes in the subtree spanned by $x_i^{\mathcal{N}}$. The second embedding captures the number of nodes on the same level as $x_i^{\mathcal{N}}$, lying to the left of it.

Analogously to the transformer, the model consists of an encoder and a decoder component (see fig. 9 left and middle). Instead of the self-attention on sequential input, the encoder calculates the attention between

all possible pairings of the leaves' $L$ and non-terminal nodes' $N$ representations. Then, the decoder calculates the cross-attention between both. Given a decoder-side query $Q \in \mathbb{R}^{t \times d}$ and the leaf and node embeddings $\mathcal{L}$ and $\mathcal{N}$, the affinity scores $A_{Q\mathcal{L}} \in \mathbb{R}^{t \times a}$ and $A_{Q\mathcal{N}} \in \mathbb{R}^{t \times b}$ are computed as follows:

$$A_{Q\mathcal{L}} = (QW_Q)(\mathcal{L}W_K)^T / \sqrt{d} \tag{3}$$

$$A_{Q\mathcal{N}} = (QW_Q)(\mathcal{N}W_K)^T / \sqrt{d} \tag{4}$$

where $W_Q$ and $W_K$ denote the trainable key and query weights.

Then, to obtain the value representation of the leaves $\mathcal{L}$, they are multiplied with the weights $\mathcal{L}W_V$. The non-terminal nodes are encoded as described above: $\bar{\mathcal{N}}' = \mathcal{V}(\mathcal{U}(\mathcal{F}(\mathcal{L}W_V, \mathcal{N}W_V, \mathcal{R}) + E), \mathcal{L}u_c)$ where $\mathcal{L}u_c$ is the learnable weight $w$ for the sum in eq. (2). Thereby the embedding of the terminating leaf for each path influences its weight in the sum.

The final cross-attention is then given as $A_Q = \text{softmax}([A_{Q\mathcal{N}}; A_{Q\mathcal{L}}])[\bar{\mathcal{N}}'; \mathcal{L}W_V]$. The subsequent FFN and Add&Norm layers work analogously to those in a regular transformer model (Vaswani et al., 2017).

## 2.2 Pointer Network for Copying Parts of the Tree

To identify the position where an axiom was applied, we construct a model that incorporates a pointing mechanism. Pointer networks have the benefit that parts of the input can be directly copied to the output. This is not possible using a regular transformer decoder architecture like it is used in the TreeTransformer which is limited to outputting tokens it has seen during training. It is beneficial for our task since it allows the model to output the node where the transformation should be applied directly, instead of having to reconstruct it from its vocabulary. Given the nature of our input data and our task, the pointer component must be equally able to point to the leaf as well as the non-terminal nodes.

In the standard transformer architecture, the queries of the attention heads of the encoder-decoder attention layers are passed through from the previous decoder layer, whereas the keys and values are obtained from the output of the encoder. Consequently, these layers capture the importance of every token in the input with regard to each token in the output sequence (Vaswani et al., 2017; Enarvi et al., 2020). Hence, the thereby emerging distribution can be interpreted as pointers to the elements of the input, i.e. those that receive a high attention are good candidates for transfer to the output.

In the TreeTransformer, the importance of each element of the tree with regard to the currently decoded token in the output sequence is read out from the cross-attention layer. It is sufficient for us to obtain the alignment scores[1] between the currently decoded element on the target side and the nodes in the input tree. To enable this, we concatenate the affinity scores $A_{Q\mathcal{L}}$ and $A_{Q\mathcal{N}}$ as defined in eqs. (3) and (4) to a new tensor $A = [A_{Q\mathcal{L}}; A_{Q\mathcal{N}}] \in \mathbb{R}^{t \times (a+b)}$. Hence, $A$ contains the alignment for each element of the tree at each decoding step $t$.

The pointer distribution $P_{point}^t(x)$ is then given by the sum over the alignment scores of each occurrence of each symbol in the input. Hence, we calculate $P_{point}^t(x)$ such that we iterate over both the input leaves $x^{\mathcal{L}}$ and non-terminal nodes $x^{\mathcal{N}}$ while summing up their respective alignment scores from $A$ as

$$P_{point}^t(x) = \sum_{i:[\mathfrak{L};\mathfrak{N}]_i = x} [A_{Q\mathcal{L}}; A_{Q\mathcal{N}}]_i^t = \sum_{i:[\mathfrak{L};\mathfrak{N}]_i = x} A_i^t. \tag{5}$$

The pointer distribution is calculated in addition to the regular distribution over the full vocabulary from the TreeTransformer's normal output. To subsequently decide if the model should copy from the input or generate new tokens, a generation probability $p_{gen} \in [0, 1]$ (See et al., 2017) is learned. In practice, to calculate $p_{gen}$ for time step $t$ a gating layer is used which is defined as

$$p_{gen}^t = \sigma \left( W[x_t; y_t] + b \right), \tag{6}$$

---

[1]Defined as the dot product between decoder-queries and encoder-keys.

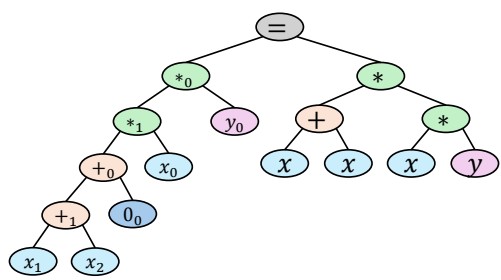

Figure 2: Visualization of our many-to-one embeddings ME. The label indicate the token of the node and the colors indicate the embedding assigned to a node. It can be seen that different tokens like $x_0$, $x_1$, and $x$ share the same embedding. Thus, they do not have to be learned independently by the network but yet are distinguishable tokens.

where $x_t$ is the embedded decoder input, $y_t$ the decoder output at time step $t$ and $W$ are learnable weights with a bias $b$ (compare the right part of fig. 9).

The final output distribution is then given by employing $p_{gen}^t$ as a soft switch between the input attention distribution and the vocabulary distribution from the decoder output. Mathematically, for each token $w$ at time step $t$ it is defined as

$$P^t(w) = p_{gen}^t P_{vocab}^t(w) + (1 - p_{gen}^t) P_{point}^t(w) \,. \tag{7}$$

Thus, a value close to one for $p_{gen}$ prefers the token predicted by the TreeTransformer decoder while a value close to zero leads to copying of a node from the input tree.

### 2.3 Many-To-One Embedding Layer

We need to be able to represent arbitrarily many occurrences of an operator, variable, or constant, for example $x_1$, $x_2$, $x_3$, in a distinguishable way. However, if we would embed them using a standard embedding layer, this would require the model to learn multiple representations of semantically identical tokens independent of each other. Since they share their meaning, we want them to also share their embedding. To this end, we introduce *many-to-one embeddings*. They enable us to include every occurrence as an individual symbol in the vocabulary while mapping all identical node types to the same shared embedding as explained in the following.

Let $\mathcal{V}$ denote a vocabulary of size $|\mathcal{V}|$ representing a set of tokens $v_0, v_1, \ldots, v_{|\mathcal{V}|}$ and $\mathcal{S}$ denote an input sequence consisting of tokens $s_1 \ldots s_n$. In the traditional setting, an embedding layer maps all occurrences of each element $s_i \in \mathcal{S}$ corresponding to the same token $v_j \in \mathcal{V}$ to the same dense embedding $e_j \in \mathbb{R}^d$, where $d$ denotes the embedding dimension.

This approach assumes all tokens $v \in \mathcal{V}$ to have a unique meaning and a large overlap of tokens between the vocabulary and the input data. However, this assumption is not justified in our setting for two reasons. If all occurrences of the same symbol would be mapped to the same token, e.g. $v_0 = $ '$*$' for the multiplication, $v_1 = $ '$+$' for the addition, the pointer could not be used to unambiguously output the position in the tree. If instead all occurrences would be added as individual embeddings—e.g. $v_0 = $ '$*_0$' for the first multiplication, $v_1 = $ '$*_1$' for the second and so on—the network would need to learn representations for all occurrences of the same symbol independently.

In addition, our model should be able to generalize to trees with a larger depth than used during training. For this it has to be able to properly deal with trees where the number of occurrences of a symbol is higher than seen during training. Therefore it is mandatory that our input embedding supports this setting and is able to generalize to this increased symbol count as well.

Consequently we define our many-to-one embeddings. Let $\mathcal{M} = \{+, *, \ldots, \sin, \cos, \ldots, 0, 1, \ldots, x, y, z\}$ describe the set of mathematical entities we use for constructing our input and let $c \in \mathbb{N}$ be a counter associated

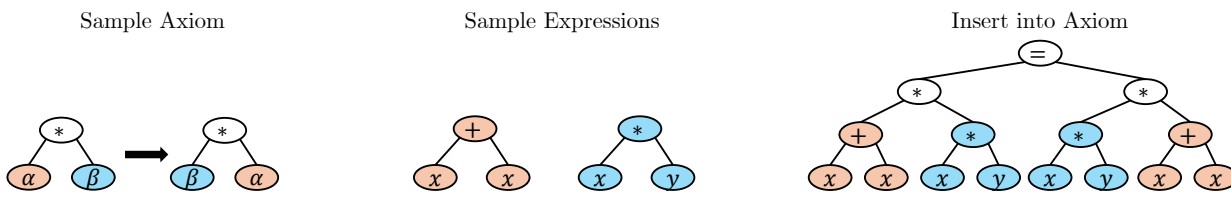

Figure 3: Generation of the initial set of equations. First an axiom is chosen. Its variables are then substituted with sampled expressions. The colors of the nodes represent the alignment between the variables of the axiom and the sampled expression.

with each token describing its index in the input (c.f. fig. 1 (left subtree)). Then, each token in our input can be considered as a tuple $v_i = (m_i, c_i), m_i \in \mathcal{M}, c_i \in \mathbb{N}$. Our many-to-one embedding layer ME has to fulfill the following properties: Given two tokens $v_a = (m_a, c_a)$, $v_b = (m_b, c_b), v_a, v_b \in \mathcal{V}$, we want $\text{ME}(v_a) = \text{ME}(v_b) \Leftrightarrow m_a = m_b$. Hence

$$\forall i \in \mathbb{N} : \text{ME}(v_k) = \text{ME}((m_k, i)) = W \cdot h(m_k)$$

with $h(m_k)$ being the one-hot vector associated with $m_k \in \mathcal{M}$ and $W \in \mathbb{R}^{|\mathcal{M}| \times d}$ is a trainable weight matrix. A Visualization of the many-to-one embeddings is given in fig. 2.

## 3 Data Set Generation

To generate data for our SETI task, we extend the generator introduced by Wankerl et al. (2023). They generate pairs of mathematical expressions with several different types of relations, such as pairs of equal expressions, pairs where one expression is the derivative of the other, or expressions which have a constant offset. For our task, we only use and modify the part of the generator which creates pairs of equal expressions (equations). Every equation can be built of up to three free variables $x, y, z \in \mathbb{R}^+$, the integer constants $-4 \ldots 4$, the real constants $\pi$ and $e$, the binary operators $+, -, \cdot, /, \hat{\ }$, and the functions ln, sin, cos, tan.

To obtain the equations, the generator works in two steps. It starts with a set of 112 axioms[2], covering mathematical subjects like polynomials, exponentiation, logarithms, and trigonometry. An axiom can be considered as an elementary mathematical rewrite rule. Exemplary axioms are $\alpha + 0 \to \alpha$, $\alpha + \beta = \beta + \alpha$, $(\alpha + \beta) + \gamma \to \alpha + (\beta + \gamma)$, or $\ln(\alpha \cdot \beta) \to \ln(\alpha) + \ln(\beta)$. Here, the arrow indicates the direction of rewriting, e.g., $\alpha + 0$ can be rewritten as $\alpha$ and $\ln(\alpha \cdot \beta)$ can be rewritten as $\ln(\alpha) + \ln(\beta)$.

**Generate Start Equations** As a first step, the generator creates a set of increasingly complex expressions by iteratively substituting already obtained expressions into the free variables of randomly sampled axioms. For example, assuming that the generator already created the expressions $(x+x)$ and $(x \cdot y)$, it can substitute them into the commutative law $(\alpha \cdot \beta \to \beta \cdot \alpha)$ yielding $(x + x) \cdot (x \cdot y) = (x \cdot y) \cdot (x + x)$. This process is depicted in fig. 3.

**Generate Axiomatic Transformations** Although the previous step is helpful to create a basic set of mathematical expressions, the data cannot be used for our SETI task itself since the transformations are not of an axiomatic nature. Therefore, as a next step, the expressions are modified according to some randomly selected matching axiom. In this step, the generator can substitute variables with parts of the expressions. For example, the generator can match the associative law $\alpha \cdot (\beta \cdot \gamma) \to (\alpha \cdot \beta) \cdot \gamma$ with the expression $(x + x) \cdot (x \cdot y)$ yielding $((x + x) \cdot x) \cdot y$. Then, the modified expression and its original form are added as a sample to the dataset: $((x + x) \cdot x) \cdot y = (x + x) \cdot (x \cdot y)$. This process is visualized in fig. 4 (upper part).

While the generator by Wankerl et al. (2023) applies this step on a random side of each equation for an undefined number of times, we instead keep track of the number of substitutions already performed and

---

[2]The full list of axioms is given in appendix B.

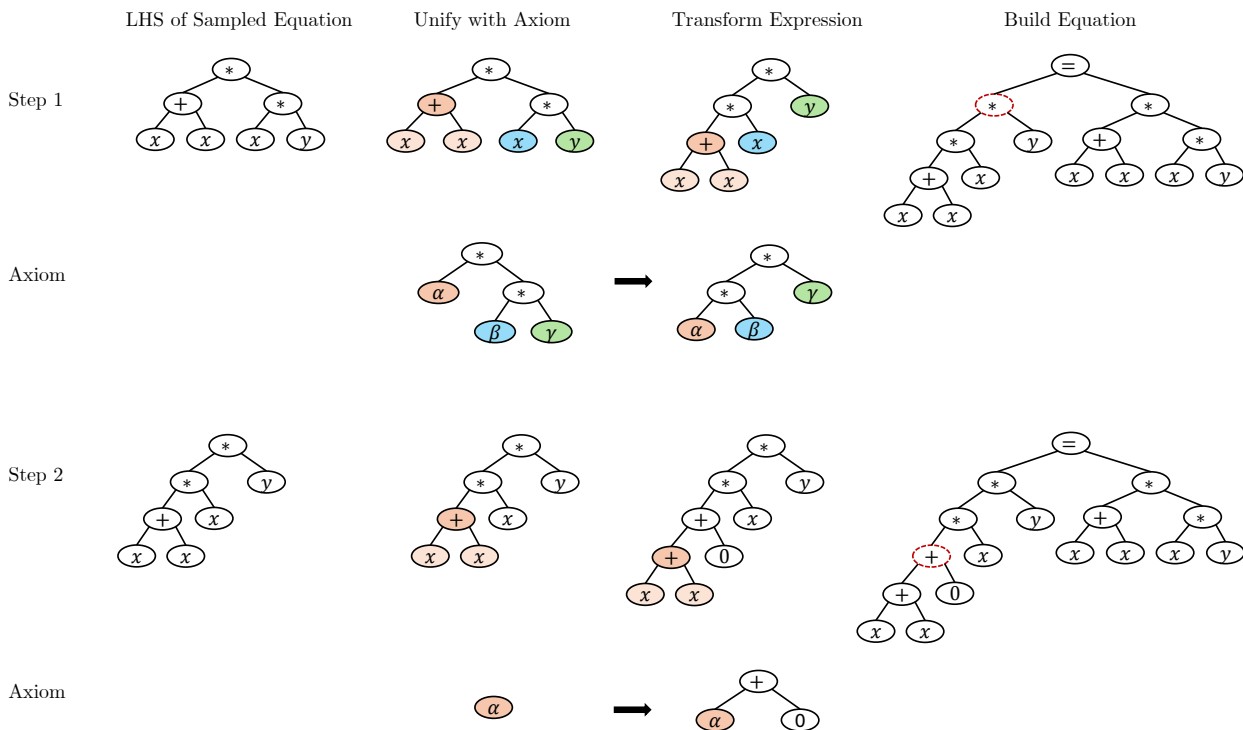

Figure 4: Visualization of generating an equation with a distance of two steps. First, the lhs of a sampled equation is extracted, unified with a random matching axiom and transformed. The original expression and its transformed form then build a new equation. By repeating this step, a second expression is generated. The rhs of the initial equation is kept and combined with the second expression to construct an equation that is two axiomatic steps apart. The colors of the nodes visualize the alignment between the expression and the variables of the axiom. The dashed red node corresponds to the root node of the modified subtree.

modify the left side of each expression only. For example, we record that $((x + x) \cdot x) \cdot y = (x + x) \cdot (x \cdot y)$ is derived in one step. Subsequently, we can apply $\alpha \to \alpha + 0$ on the expression, yielding an equation that is transformed in two steps: $(((x + x) + 0) \cdot x) \cdot y = (x + x) \cdot (x \cdot y)$. If an axiom matches the expression at multiple positions, a random node is selected as the root node for applying the axiom. This process is visualized in fig. 4 (lower part).

The generator as described so far might generate redundant samples where an axiomatic transformation is immediately reversed in the next step. Above example might be reversed to $((x + x) \cdot x) \cdot y = (x + x) \cdot (x \cdot y)$ by applying the reverse axiom from before $(\alpha + 0 \to \alpha)$. To prevent the generator from creating such loops, we forbid the usage of axioms which would reverse the previously generated expression. Furthermore, to ensure a more uniform distribution of the axioms, we count how many times each matching axiom has been used in the pairs generated so far, and use the inverse of that count as probabilities for sampling.

**Check Validity of Transformations**   The mere application of axioms described so far can create mathematically invalid expressions. For example, $(\frac{1}{\ln(1)})^{-1}$ could be generated, but since $\ln(1) = 0$, this expression is undefined. To avoid the inclusion of such samples in the dataset, all expressions are checked using Sympy (Meurer et al., 2017). If variables appear in the expression, they are assumed to be positive real numbers. All expressions that Sympy evaluates to NaN or infinity are discarded and thus do not appear in the dataset.

**Make all Tokens Uniquely Identifiable**   To make the occurrences of the mathematical tokens inside an expression uniquely identifiable, we enumerate the tokens of the left side of each equation by adding a counter to each type of token. Above example is therefore represented as $(((x_1 +_1 x_2) +_0 0_0) \cdot_1 x_0) \cdot_0 y_0 = (x + x) \cdot (x \cdot y)$.

**Build Samples for the SETI task** Every sample can be described as a 3-tuple $(\text{lhs}_i = \text{rhs}_i, a_i, p_i)$. Here, $\text{lhs}_i = \text{rhs}_i$ denotes a mathematical equation and $a_i$ denotes an axiom that, when applied at position $p_i$, describes a step for transforming $\text{lhs}_i$ towards $\text{rhs}_i$. The position $p_i$ matches the root node of the subtree inside the parse tree of $\text{lhs}_i$ that should be modified when applying the axiom $a_i$ (c.f. fig. 1). However, every sample describes just one possible step and not the whole chain of transformations between the left and the right side. Above example thus corresponds to the following tuple: $((((x_1 +_1 x_2) +_0 0_0) \cdot_1 x_0) \cdot_0 y_0 = (x+x) \cdot (x \cdot y), \alpha + 0 \to \alpha, +_0)$.

We create about 8.5 million samples for training, 400,000 samples for validation and 14,000 samples for testing. In the training set, we set the maximum distance between two expressions to five, meaning that the left side of all equations can be transformed into the corresponding right side in at most five steps. Moreover, in our training set we only include equations whose parse trees have a maximum depth of 7. Detailed statistics of the data can be found in appendix B.

## 4 Experiments

In this section, we describe our experimental setup including baselines for comparison, how the target sequence is structured, and our used hyperparameters.

### 4.1 Baselines

We employ three additional baselines: a transformer (Vaswani et al., 2017), a pointer network by Enarvi et al. (2020) to which we refer as *SeqPointer* and a recurrent seq2seq model that makes use of an LSTM (Hochreiter & Schmidhuber, 1997) encoder and decoder with Bahdanau attention (Bahdanau et al., 2015). We add the last model to allow an ancillary comparison with a recurrent neural network architecture besides all the other transformer-based models. For brevity, we refer to this model as *LSTM*.

All three models receive the prefix representation of each input equation. We experiment with different tokenization strategies. As a first variant, we represent the mathematical symbol and its index as one token. For example, the equation $\ln(1)/1 = 0$ is tokenized to `[=, /_0, ln_0, 1_0, 1_1, 0]`.

Although this tokenization is equal to the tokenization for TreePointerNet, it has the disadvantage that the tokens corresponding to high indices appear much less frequent in the dataset. To alleviate this problem, we also experimented with a second tokenization strategy, namely representing the index as a separate token. Above example would then be represented as `[=, /, 0, ln, 0, 1, 0, 1, 1, 0]`. However, in our experiments it yielded inconclusive results compared to the first tokenization. Since no tokenization is superior compared to the other, we stick to the results obtained with the first tokenization for our experiments because it equals the tokenization for TreePointerNet. We present the results obtained with the second tokenization in appendix E.

The prefix notation has the advantage that it is more compact than the human-readable infix notation since it does not require parentheses. It has also been used by various previous research employing sequential models for symbolic mathematics (Lample & Charton, 2019; D'Ascoli et al., 2022; Wankerl et al., 2023).

We also experimented with LLMs as a baseline for the SETI task. However, in our experiments it was not solvable using the tested models and hence we do not include it as an additional baseline in our experiments. Details about our experiments with LLMs can be found in appendix F.

### 4.2 Modeling the Target Sequence

All target sequences for all models consists of two tokens: a class token for the applied axiom and the root node of the modified subtree in the expression tree. As commonly done when training seq2seq models, we explicitly terminate each sequence with an end-of-sequence token. An exemplary target sequence might hence be: `ax_1, y_2, EOS`.

### 4.3 Evaluation Strategy

The input to the models is an equation that is either given as a parse tree (c.f. fig. 1) or as a sequence as described in section 4.1. The output of the models are two tokens (axiom and position) as described in section 4.2. To find a sequence of axiomatic steps for transforming the left side of an equation into the right side, we repeat below procedure for up to $n = 5$ step. We use greedy decoding, i.e. in every step, we chose the most likely pair of axiom and position.

Let $\hat{a}_i$ denote the predicted axiom and $\hat{p}_i$ denote the predicted position within equation $\text{lhs}_i = \text{rhs}_i$. We use a custom rule-based algorithm to verify if $\hat{a}_i$ can be applied at node $\hat{p}_i$. Precisely, we check if $\hat{p}_i$ exists within $\text{lhs}_i$ and if it spans a subtree where $\hat{a}_i$ matches. If this is not the case, we terminate the process and count the equation as unsolved. Otherwise, we construct $\text{lhs}_i^*$ by applying the axiom in $\text{lhs}_i$. Then, we check if $\text{lhs}_i^*$ and $\text{rhs}_i$ are syntactically identical. If this is the case we count the equation as solved. Otherwise, we feed it into the neural network again, obtaining the next transformation step. If we did not obtain a syntactically equal equation after $n$ steps, it also counts as unsolved.

### 4.4 Ablation Study and Generalization Ability

We perform an ablation study on our TreePointerNet. Precisely, we remove the pointer-copy mechanism and the many-to-one embedding, respectively, to quantify their benefit for the task at hand. Thus, we retrain the model without one of these components while leaving all other hyperparameters unchanged to rule out their potential influence on the obtained results.

In addition, we test the model on deeper trees (depth 8–12) than seen during training (depth up to 7). Furthermore, we test on equations that require more steps to transform (6–9 steps) than those seen during training (1–5 steps). In doing so, we explicitly test how well our model generalizes to trees which lie outside the training distribution and represent mathematical expressions of higher compositionality.

### 4.5 Implementation and Training

All our experiments are implemented using the Fairseq (Ott et al., 2019) toolkit. We train all models using the Adam optimizer (Kingma & Ba, 2015) on batches of 16 samples for up to 100 epochs or until the loss stagnates or deteriorates over a period of 10 epochs. We ran a hyper-parameter search using the Optuna framework (Akiba et al., 2019), exploring 100 configurations each for our model and the baselines. Our search space included the number of layers in the encoder and decoder, the number of attention heads used per layer, the number of attention heads used for pointing (where applicable), the size of embeddings and the hidden representations in the encoder and decoder, the dropout rates and the learning rate of the optimizer. The exact parameters and search spaces are given in appendix C.

Due to the hyperparameter optimization, our TreePointerNet has only 2,649,901 trainable parameters. Thus, this model requires a much lower number of parameters for achieving optimal results compared to SeqPointer and transformer (22,768,644) and LSTM (6,496,256). Hence, it is particularly efficient with regard to the number of parameters, as it is very well tailored to our SETI task. On a modern multi-core CPU, evaluating TreePointerNet averages to 21 equations per second.

## 5 Results and Analysis

In this section we present and discuss our obtained results. Figures 5 to 8 show the average accuracy ($\pm$ standard deviation over five model runs, each time initialized with a different random seed), i.e. the fraction of correctly transformed equations grouped by the number of required transformation steps. The detailed numeric results are given in appendix D.

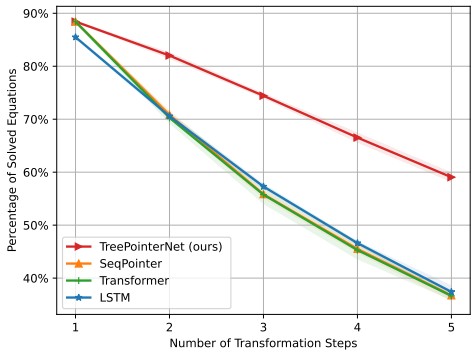

Figure 5: Results for all Models on Equations of up to 5 Required Transformation Steps

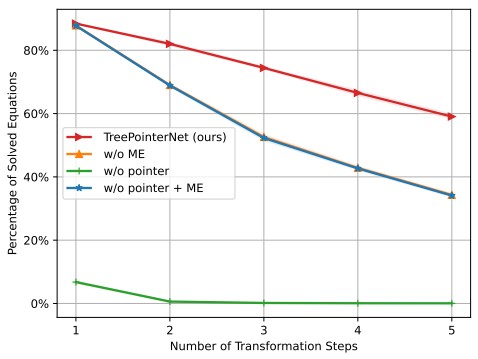

Figure 6: Ablation Study on TreePointerNet

## 5.1 Model Performance

As can be seen in fig. 5, TreePointerNet outperforms all baselines by a large extent, achieving an overall accuracy of 74.09%, whereas the SeqPointer only achieves 59.47% and the transformer is even weaker with 59.28%. The LSTM achieves an accuracy of 59.47%. All models prove to be stable (stdev below 2%).

Considering the equations where only one transformation step is required, it is noticeable that the performance of all models is very similar. Here, TreePointerNet is on par with a transformer, both reaching an accuracy of 88.43%, and SeqPointer (88.41%). Only the LSTM (85.49%) performs slightly worse. It is therefore standing to reason that, given an adequate number of trainable parameters, all tested neural architectures can comparably recognize elementary axiomatic differences between two mathematical expressions.

When analyzing the equations requiring multiple transformation steps, a clear decline in the accuracy of all models can be observed with each additional transformation step. Yet, the TreePointerNet outperforms the baselines to a large extend since it looses much less accuracy than all baselines. For example, when going from one to two transformation steps, the best baseline (SeqPointer) already looses 17.58 percentage points, whereas TreePointerNet only looses 6.41 percentage points. Thus, TreePointerNet preserves an accuracy of 82.02% on theses equations, whereas SeqPointer reaches only 70.83%. This trend continues. On the equations requiring five steps, TreePointerNet achieves a total accuracy of 59.05% which is only 29.38 percentage points below the accuracy on equations of one step. However, the other models loose on average 50.51 percentage points, with the LSTM being the most robust model achieving an accuracy of 37.4%.

Predicting transformation steps for equations which are more than one step apart require the networks to identify more complex mathematical patterns than the mere axiomatic differences that are equally well recognized by all models. Understandably, models which capture those patterns with less certainty quickly loose more overall accuracy with each step, since each step corresponds to an independent classification.

Summarizing above results, our TreePointerNet model clearly outperforms the baselines and proves to be stable and efficient. At the same time it is noticeably more robust to changes in the number of required transformation steps where it can generalize with significantly smaller performance loss. For illustrative purposes, we present a few exemplary equations as they were transformed by TreePointerNet in appendix G.

## 5.2 Ablation Study

We further analyze the influence of the many-to-one embedding and the copy-pointer on the TreePointerNet's performance by removing one of these components a time. The results are presented in fig. 6.

Removing the many-to-one embeddings, the model's performance drops to an average accuracy of 57.31% making it comparable to the performance of the baseline models. In particular, the model reveals the same

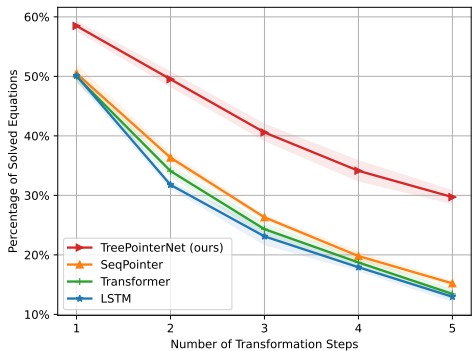
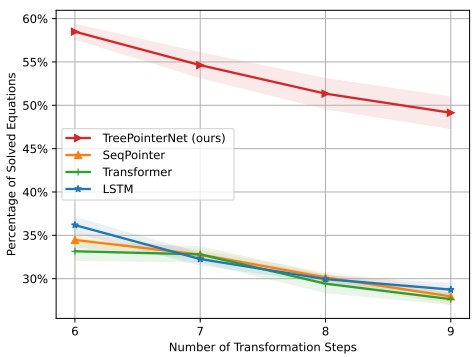

Figure 7: Robustness Study for all Models on Equations of Deeper Parse Trees

Figure 8: Robustness Study for all Models on Equations of up to 9 Transformation Steps

decline in accuracy on the equations with more transformation steps. This supports our motivation from section 2.3 for introducing the many-to-one embeddings.

First, without these embeddings the model now has to learn a single representation for each occurrence of a mathematical token instead of sharing them. This likely decreases the performance as there is less training data available to learn the mathematical meaning of each symbol. Second, several symbols are potentially mapped to an untrained embedding since they did not occur in the training data. Hence, the model can not understand their semantics which leads to a stronger decline in prediction performance.

Removing the pointer component causes the performance of the model to drop to an average accuracy of 1.53%. This behavior can be understood when considering the many-to-one embeddings. Since all occurrences of the same symbol are mapped to the same embedding, the generative part of the network can no longer distinguish between them. Hence the model can not identify the position where the axiom should be applied.

The pointer component which directly extracts them from the alignment scores between the output tokens and the nodes (cf. section 2.2) is necessary to benefit from using the many-to-one embeddings. When removing both components at the same time, the average accuracy drops to 57.14%, marginally worse than when only removing the many-to-one embeddings. We conclude that having a pointer component is slightly beneficial for our task, even without our many-to-one embeddings.

### 5.3 Robustness on Data of Higher Complexity

We perform two further experiments to evaluate the robustness of our model on data of unseen complexity. First, we increase the complexity of the equations themselves. In line with previous research (Arabshahi et al., 2019; Mali et al., 2021; Wankerl et al., 2023), we use the depth of the parse tree as the measure of complexity. The results are presented in fig. 7. All models clearly loose accuracy on this data, but on average TreePointerNet (42.48%) still outperforms all baselines (27.18% - 29.63%). Beyond that, this experiment reveals a similar overall trend as it can be observed on the shallow trees in that the performance of all models declines with the number of transformation steps. Yet, in contrary to the latter, TreePointerNet already outperforms the baselines on the equations of one transformation step by at least 7.99 percentage points. Thus, TreePointerNet identifies the mathematical patterns in data of unseen complexity more robustly than the standard models.

Second, we evaluate the models on equations which require more transformation steps than seen during training. The results are shown in fig. 8 and seamlessly connect to the results presented in fig. 5. Therefore, we observe a continuation of the previous trend, with TreePointerNet still outperforming the baselines. On our tested maximum of nine required transformation steps, TreePointerNet still reaches an accuracy of 49.14% which is at least 20.4 percentage points above the baselines.

Table 1: Accuracy by Depth of Equation Tree and Level of Application of Axiom

| Depth of Tree | Level of Transformation Root | | | | | | Average |
|---|---|---|---|---|---|---|---|
| | 1 | 2 | 3 | 4 | 5 | 6 | |
| 3 | 97.0% | 90.9% | - | - | - | - | 94.0% |
| 4 | 98.3% | 96.4% | 92.2% | - | - | - | 95.6% |
| 5 | 98.4% | 91.8% | 91.8% | 90.5% | - | - | 93.1% |
| 6 | 98.6% | 91.0% | 82.7% | 84.1% | 84.3% | - | 88.1% |
| 7 | 99.5% | 88.4% | 81.5% | 76.5% | 78.1% | 77.7% | 83.6% |

Table 2: Most Frequently Misclassified Axioms and their Counterparts

| Axiom | Confused with |
|---|---|
| $(\alpha \cdot \beta)^\gamma \rightarrow (\alpha^\gamma) \cdot (\beta^\gamma)$ | $(\alpha \cdot \beta) \cdot \gamma \rightarrow \alpha \cdot (\beta \cdot \gamma)$ |
| $\cos(\alpha) \rightarrow \cos(-1 \cdot \alpha)$ | $0 \rightarrow \alpha \cdot 0$ |
| $\alpha \cdot \beta^{-1} \rightarrow \alpha/\beta$ | $(\alpha \cdot \gamma)/\beta \rightarrow \alpha/(\beta/\gamma)$ |
| $(\alpha + \beta) + \gamma \rightarrow \alpha + (\beta + \gamma)$ | $(\alpha \cdot \beta) \cdot \gamma \rightarrow \alpha \cdot (\beta \cdot \gamma)$ |
| $\alpha - \beta \rightarrow \alpha + (-1 \cdot \beta)$ | $\cos(\pi - \alpha) \rightarrow -1 \cdot \cos(\alpha)$ |
| $\alpha \cdot \beta \rightarrow \beta \cdot \alpha$ | $\alpha + \beta \rightarrow \beta + \alpha$ |

## 5.4 Interpreting the Model's Predictions

To make the results of our model more illustrative, we want to discuss prominent patterns in our model's predictions. Precisely, we want to identify axioms which are commonly confused and analyze the influence of the depth of the whole equation tree as well as the level of the root node where the axiom has to be applied. However, due to the nature of the SETI task, unambiguous labels only exist for the subset of equations that can be derived in one step. Thus, the analysis in this section is performed on this subset.

We start with evaluating the performance of TreePointerNet with regard to the depth of the tree and the level of the root node of the applied transformation. The results are presented in table 1. We find an overall decline in the accuracy the deeper the input tree. On shallow trees with a depth of 3-5, the transformations are identified with an accuracy of at least 93.1%. The accuracy drops to 88.1% on trees of depth 6 and 83.6% on trees of depth 7. However, this result seems natural since deeper trees generally correspond to longer and more complex equations making the correct transformation harder to identify.

Considering the level of the root node of the transformation within the equation, by tendency we observe the accuracy to decline the deeper the level of the root node is within the tree. This is true independent of the overall depth of the tree. Descriptively, the deeper the level of the transformation the closer it is to the leaf nodes. Thus, a transformation applied close to the leaf nodes corresponds to a more local change within the equation. While transformations which occur directly under the root node (level 1) are recognized with an accuracy of at least 97%, those which are on the deepest possible level are only recognized with an accuracy between 92.2% (tree of depth 3) and 77.7% (trees of depth 7). So overall the deeper, i.e., more local and hence smaller, the change is, the worse it is recognized by the model.

Finally, we analyze the model's ability to recognize the applied axiom. We found that from the 112 axioms we use in our data set, the vast majority is classified correctly in nearly all cases. Overall, there are only six axioms where the model has larger difficulties and which could only be identified correctly in 70%–90% of all cases. They are listed in table 2 together with the axiom they are most likely confused with in descending order, i.e., the topmost axiom is the one which is most likely misclassified. Keep in mind that only the left side of the axioms are matched with the expression tree.

The misclassified axioms reveal the prominent patterns that they always match very similar trees like the axioms they are confused with. The axioms $\cos(\alpha) \rightarrow \cos(-1 \cdot \alpha)$ that is confused with $0 \rightarrow \alpha \cdot 0$ and $\alpha - \beta \rightarrow \alpha + (-1 \cdot \beta)$ that is confused with $\cos(\pi - \alpha) \rightarrow -1 \cdot \cos(\alpha)$ are structurally identical apart from the

cos function (corresponding to a unary node). All other axioms correspond to structurally identical trees which only differ in the operators or constants. For example $\alpha \cdot \beta \rightarrow \beta \cdot \alpha$ and $\alpha + \beta \rightarrow \beta + \alpha$ only differ in the binary operator node (multiplication vs addition). Similarly, $(\alpha \cdot \beta)^{\gamma} \rightarrow (\alpha^{\gamma}) \cdot (\beta^{\gamma})$ and $(\alpha \cdot \beta) \cdot \gamma \rightarrow \alpha \cdot (\beta \cdot \gamma)$ only differ in the rightmost binary operator node (power vs multiplication). We therefore conclude that TreePointerNet has minor difficulties when it comes to distinguishing structurally very similar trees and trees which essentially only differ in the specific values of the nodes.

# 6 Related Work

In this work, we introduce the SETI task. To the best of our knowledge, there exists no closely related work that addresses this kind of problem. Yet, there is loosely connected work we want to discuss in this section.

**Pointer Networks** Pointer networks describe a class of neural network architectures which learn to predict the conditional probability over positions or indices of the input sequence instead of tokens from e.g. a fixed vocabulary. They were originally introduced to approximate NP-hard geometrical problems by Vinyals et al. (2015). The approach was adopted for various tasks where the extraction of parts of the input is required or helpful, e.g. summarization (Gu et al., 2016; Miao & Blunsom, 2016; See et al., 2017; Enarvi et al., 2020; Gulcehre et al., 2016), sentiment analysis (Yan et al., 2021; Pfister et al., 2022), relation extraction (Nayak & Ng, 2020) and signal analysis (Moussa et al., 2023).

Most of above works introduce pure pointer networks. Technically, these networks use an attention distribution to identify relevant sections in their input. Therefore, they can be used to copy or indicate parts of the input but they cannot generate new tokens. Thus, they are not useful for our research as they cannot generate a token for indicating the applied axiom.

However, there also exist pointer-generator networks (See et al., 2017; Enarvi et al., 2020). These models are able to copy tokens from the input sequence to the output sequence, along with generating new tokens. They are implemented by augmenting the decoder of a regular seq2seq model with a pointer network and a gating layer that switches between the two components. While See et al. (2017) implement their network based on an LSTM encoder-decoder architecture, Enarvi et al. (2020) augments a transformer encoder-decoder architecture with a pointer network. Our research builds on this idea and introduces a model that is not limited to pointing on sequences but can process tree-structured input.

**Deep Learning on Tree-Structured Input** Various neural architectures have been designed to make use of tree-structured input. Socher et al. (2013) use neural networks to process trees recursively in a bottom-up manner. Starting from the leave nodes, for each node a representation is calculated based on the value of its children. This approach is roughly comparable to RNNs on sequences where the hidden state is updated based on previous elements of the sequence. Tai et al. (2015) build on this idea and incorporate gating mechanisms into the recursive architecture, comparable to an LSTM for sequences. Arabshahi et al. (2019) further extend the model and add a distinct memory to each node in the form of a differentiable stack. All these models require recursive function calls and are therefore difficult to parallelize for training on GPUs.

Alternative to the recursive models, Mou et al. (2016) developed a convolutional architecture for trees. Here, learnable filters merge each node with its children, allowing the network to learn a representation of the tree based on local relationships. Furthermore, Bai et al. (2021) incorporate trees into pretrained transformer encoders by masking the self-attention in accordance with the structure of the input tree.

However, all above models were designed for classification tasks and are unable to generate output sequences. Thus, in our setting they could only be used to predict an axiom or a position, but not both. Furthermore, apart from Bai et al. (2021), none of the models incorporate any form of attention mechanism which is needed to calculate a pointer distribution. Hence, in our setting they could not directly extract the position from the input tree. To the best of our knowledge, the TreeTransformer (Nguyen et al., 2020) is the only architecture that can map a tree to an arbitrary sequence and incorporates an attention mechanism. Thus, we decided to use it as the base architecture for our TreePointerNet and extend it with the features we need for our SETI task, namely a pointer component and our custom many-to-one embeddings.

**Deep Learning for Symbolic Mathematics**   Recent research explored the capabilities of deep learning architectures on various tasks from the realm of symbolic mathematics. Arabshahi et al. (2019) aimed to explore if neural networks can learn the equivalence of mathematical expressions. To this end, they introduced a synthetically generated data set (Arabshahi et al., 2018) for mathematical equations from the field of elementary algebra and trigonometry. Mali et al. (2021) improved the state on this dataset by developing more complex, higher-order models. Wankerl et al. (2021; 2023) introduced new datasets that overcame various biases (Davis, 2021) and added more mathematical relations and neural architectures.

Apart from classifying mathematical relations, seq2seq models were used for various symbolic and arithmetic math problems (Saxton et al., 2018), for example integration and differential equations (Lample & Charton, 2019) and linear algebra (Charton, 2022). Other researchers combine natural language with formal mathematics to solve word problems (Hendrycks et al., 2021; Azerbayev et al., 2024; Yu et al., 2024). All these models are trained end-to-end to output the solution of the given input expression. Here, only the final answer of the LLM is evaluated making it possible that intermediate steps are incorrect. In addition, it could be shown that transformers do not reliably generalize to numerical tasks like addition or multiplication with specific numbers (Welleck et al., 2022) and frequently make mistakes on non-axiomatic transformations as required for calculating integrals. Hence, we focus on a mostly symbolic task and include specific numbers only peripherally. Furthermore, we want to output single steps where each step can be verified.

Finally, there is recent research on proving mathematical theorems (Azerbayev et al., 2024; Song et al., 2023) by combining LLMs with external prove assistants like Lean (De Moura et al., 2015) using a framework like Welleck (2023). Here, the LLM is given the current state of the theorem prover and its goal is to propose or select a next prove step, i.e. axiom. However, this differs from our approach, as we use the neural network to simultaneously predict both the axiom and the position at which it is applied. Moreover, their focus lies on verifying or disproving the equality of given expression rather than finding paths to transform one expression into another equal expression. Hence, we did not integrate our research into this setting.

# 7   Future Applications and Extensions of our Model and Data

The model and dataset we introduced can be beneficial for various further applications and research. In this section, we provide a brief outline of potential future work that could benefit from our data or model.

In an educational setting, the model could be used to help students by providing hints and intermediate steps in an interactive teaching tool. Moreover, on a learning platform where students solve exercises, the model could help to identify individual knowledge gaps and enhance recommender systems in finding suitable exercises. More precisely, since the model can identify the axioms needed to solve an equation, it could also be used to identify potentially unknown axioms of students or groups of similar students. Then, students could be redirected to appropriate reading material or further exercises which advances their personal learning experience.

Furthermore, the dataset or generator could also be helpful to improve research on word problems (i.e. mathematical problems where the problem is partly formulated using natural language). So far, we input an equation to generate a step-by-step solution, while other research focusing on word problems (c.f. section 6) inputs texts. Yet, in future research both approaches could be combined to allow similarly fine-grained solutions to word problems.

Moreover, our research could be extended to other domains of mathematics. The axioms we use in this work cover polynomials, logarithms, exponentiation, and trigonometry. Thus, we are mainly considering mathematics on a level as it is taught in high-school or early college. Furthermore, so far our data contains only a few constants and variables are assumed to be positive real numbers. Yet this does not impose a conceptual limitation on the architecture of TreePointerNet. Future work could therefore extend the set of axioms to more advanced branches of mathematics and use TreePointerNet on it.

In addition, the setting could also be extended to other mathematical tasks beside finding steps to show equality. For example, seq2seq models were proposed to calculate integrals (Lample & Charton, 2019), outputting the final solution in one step. Therefore, one could investigate if our model can be used to

generate step-by-step calculations for finding integrals or derivatives in a similarly fine-grained way we do it for the equivalence in this work.

## 8 Conclusion

In this work, we introduce the new Stepwise Equation Transformation Identification (SETI) task of fine-grained prediction for axiomatic mathematical transformations. Given two equivalent mathematical expressions, the task is to iteratively predict a sequence of steps for transforming one expression into the other. Each step consists of both the axiom and the position where it must be applied to transfer the first expression towards the second. To this end, we generate a new equation data set consisting of pairs of mathematical equivalent expressions represented by expression trees. We then solve the task using TreePointerNet, a new architecture combining a pointer generator network with a hierarchical-accumulation model for tree-structured input and a novel embedding strategy. We show that our model is able to consistently outperform even strong baselines and conclude that our network benefits from the ability to make use of the inherent hierarchical structure of expression trees, the pointer component and the many-to-one embeddings.

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

## A   Further Details of the TreePointerNet Architecture

### A.1   Visualization of the TreePointerNet Architecture

Figure 9 provides a visualization of our TreePointerNet as described in section 2. The left and middle parts show the TreeTransformer architecture introduced in section 2.1. The equations are input as trees (bottom left). The right part shows the pointer generator network (section 2.2) which outputs a mix of generated and copied tokens sampled from the vocabulary distribution and the pointer distribution (weighted by $p_{\text{gen}}$). The many-to-one embeddings (section 2.3) are input to both the encoder and the decoder part of the network as visualized in fig. 9 (left and middle, bottom).

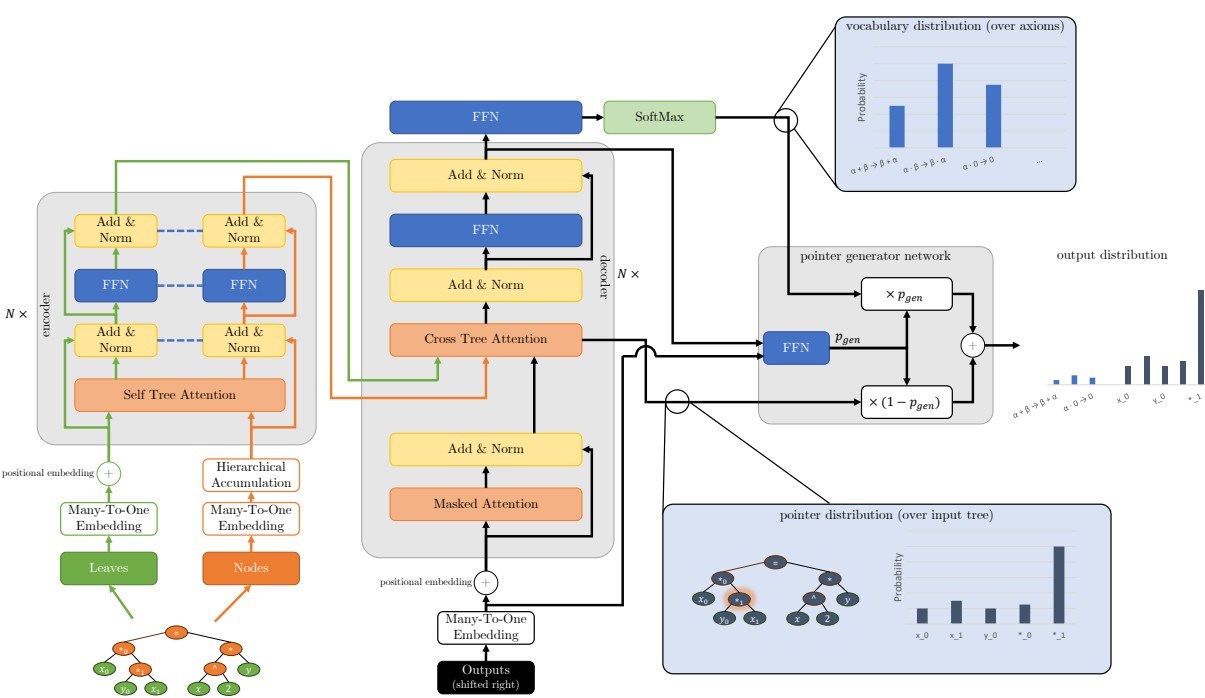

Figure 9: Overview over our architecture. Both the transformer encoder (left) and decoder (middle) make use of the tree-structured attention. From the last decoder layer the decoder output as well as the cross-attention is fed together with the embedded input tokens to the pointer network (right).

### A.2   Complexity of TreePointerNet

With regard to the computational complexity, there is no difference between a transformer and TreePointerNet. Let $n$ denote the length of the input sequence (for sequential models like transformers) or the size of the input tree (for tree-based models like TreePointerNet). The transformer requires $\mathcal{O}(n^2 \cdot d)$ steps per self-attention layer, where $d$ is the size of the learned representations (Vaswani et al., 2017). Assuming that $d$ is set to a fixed value, the complexity is dominated by the length of the sequence, and thus $\mathcal{O}(n^2)$. This is equally true for the TreeTransformer as proven in Nguyen et al. (2020).

Our many-to-one embeddings do not add any overhead with regard to the computational complexity. Each many-to-one embedding can be computed in $\mathcal{O}(1)$ using a hash-table for mapping an input token to its embedding. Since for all input tokens an embedding has to be calculated, this requires $\mathcal{O}(n)$ steps in total. Furthermore, the pointer distribution of TreePointerNet merely requires a summation over the already computed attention scores of all occurrences of every input token. Thus, it can be computed in $\mathcal{O}(n)$ (see

Table 3: Comparison of TreeTransformer with an Equal-Sized Transformer

| Model | Number of Transformation Steps | | | | | Average |
|---|---|---|---|---|---|---|
| | 1 | 2 | 3 | 4 | 5 | |
| TreeTransformer | $87.76\%_{\pm 0.19\%}$ | $\mathbf{68.87\%}_{\pm 0.42\%}$ | $\mathbf{52.29\%}_{\pm 0.79\%}$ | $\mathbf{42.67\%}_{\pm 0.51\%}$ | $\mathbf{34.10\%}_{\pm 0.42\%}$ | $\mathbf{57.14\%}$ |
| Transformer | $\mathbf{88.67\%}_{\pm 0.05\%}$ | $65.50\%_{\pm 0.45\%}$ | $49.05\%_{\pm 0.36\%}$ | $39.22\%_{\pm 0.64\%}$ | $31.88\%_{\pm 0.73\%}$ | $54.86\%$ |

eq. (5) in section 2.2). Concluding, TreePointerNet has the same complexity as the above mentioned baseline models, namely $\mathcal{O}(n^2)$.

### A.3 Parameter Efficiency of TreePointerNet

In section 2, we hypothesized that a model explicitly using the tree structure of mathematical equations can achieve better results with less parameters than a model which has to deduce the grammar of mathematics from sequential input. To confirm this claim, we experimented with a transformer of the same size as our TreePointerNet and compared it with the performance of the TreeTransformer[3]. Thus, the transformer in this experiment has only 2,649,901 trainable parameters, roughly $\frac{1}{9}$ of the parameters of the transformer we use in all other experiments (22,768,644).

Table 3 presents the results of this experiment. Although the transformer is able to learn equations which require only one transformation step with higher accuracy, it proves to be less capable to generalize on equations which require more transformation steps. Overall, the TreeTransformer outperforms the transformer by 2.28 percentage points. We conclude that using tree-structured input is beneficial for our SETI task and therefore the TreeTransformer functions as a suitable basis for our TreePointerNet.

## B Dataset Axioms and Statistics

### B.1 Axioms

Table 4 lists all 112 axioms we used for creating our datasets. The axioms are taken from Wankerl et al. (2023). For brevity, the axioms are listed as equivalences. However, in practice every equivalence corresponds to two axioms, one for each direction of application. For example, $\alpha/\beta \leftrightarrow \alpha \cdot \beta^{-1}$ represents $\alpha/\beta \to \alpha \cdot \beta^{-1}$ and $\alpha \cdot \beta^{-1} \to \alpha/\beta$, where the arrow indicates the direction of application, i.e. the left side is replaced with the right side.

The only exceptions are the commutative law for the addition ($\alpha + \beta \to \beta + \alpha$) and the multiplication ($\alpha \cdot \beta \to \beta \cdot \alpha$). These two axioms are not applied from right-to-left because the result would be indistinguishable from applying them left-to-right.

### B.2 Dataset Statistics

Our training dataset contains around about 8.5 million samples consisting of trees having a depth up to 7. The test dataset for our main experiments (cf. section 5.1) contains $13,641$ samples and comes from the same distribution as the training data. In table 5 we show the distribution of depth of the trees and the level of the transformation root node within the trees of our test dataset.

It is clearly visible that the number of samples in the dataset increases with the depth of the tree. This is caused by the fact that more possible equations exist the deeper the tree. Especially in the realm of very shallow trees of depth 3–4, the number of possible equations is very limited and hence we decided not to balance the dataset with regard to the depth.

Considering the level of the root node of the transformation, we observe a light tendency towards the nodes on a middle level and below. This is standing to reason since (on average) more nodes exist at the deeper

---

[3]The TreeTransformer corresponds to our model without its pointer component and the many-to-one embeddings.

Table 4: Axioms Used to Create the Data Sets

| Arithmetic | Exponential and Logarithm |
|---|---|
| $(\alpha + \beta) + \gamma \leftrightarrow \alpha + (\beta + \gamma)$ | $\alpha^1 \leftrightarrow \alpha$ |
| $\alpha + \beta \rightarrow \beta + \alpha$ | $\alpha^0 \leftrightarrow 1$ |
| $(\alpha \cdot \beta) \cdot \gamma \leftrightarrow \alpha \cdot (\beta \cdot \gamma)$ | $\alpha^{(\beta+\gamma)} \leftrightarrow (\alpha^\beta) \cdot (\alpha^\gamma)$ |
| $\alpha \cdot \beta \rightarrow \beta \cdot \alpha$ | $(\alpha \cdot \beta)^\gamma \leftrightarrow (\alpha^\gamma) \cdot (\beta^\gamma)$ |
| $\alpha \cdot (\beta + \gamma) \leftrightarrow \alpha \cdot \beta + \alpha \cdot \gamma$ | $(\alpha^\beta)^\gamma \leftrightarrow \alpha^{(\beta \cdot \gamma)}$ |
| $\alpha + 0 \leftrightarrow \alpha$ | $1^{-1} \leftrightarrow 1$ |
| $\alpha \cdot 0 \leftrightarrow 0$ | $\alpha/\beta \leftrightarrow \alpha \cdot \beta^{-1}$ |
| $\alpha \cdot 1 \leftrightarrow \alpha$ | $\alpha/\beta \leftrightarrow (\beta/\alpha)^{-1}$ |
| $1 + 1 \leftrightarrow 2$ | $\alpha/(\beta/\gamma) \leftrightarrow (\alpha \cdot \gamma)/\beta$ |
| $2 + 1 \leftrightarrow 3$ | $\ln(\alpha^\beta) \leftrightarrow \beta \cdot \ln(\alpha)$ |
| $3 + 1 \leftrightarrow 4$ | $\ln(\alpha \cdot \beta) \leftrightarrow \ln(\alpha) + \ln(\beta)$ |
| $-1 \cdot 1 \leftrightarrow -1$ | $\ln(1) \leftrightarrow 0$ |
| $-2 \leftrightarrow -1 \cdot 2$ | $\ln(e) \leftrightarrow 1$ |
| $-3 \leftrightarrow -1 \cdot 3$ | $e^{\ln(\alpha)} \leftrightarrow \alpha$ |
| $-4 \leftrightarrow -1 \cdot 4$ | |
| $-1 \cdot -1 \leftrightarrow 1$ | |
| $\alpha - \beta \leftrightarrow \alpha + (-1 \cdot \beta)$ | |
| $\alpha - \alpha \leftrightarrow 0$ | |

| Trigonometrical | |
|---|---|
| $\tan(\alpha - \beta) \leftrightarrow (\tan(\alpha) - \tan(\beta))/(1 + \tan(\alpha) \cdot \tan(\beta))$ | |
| $\tan(\alpha + \beta) \leftrightarrow (\tan(\alpha) + \tan(\beta))/(1 - \tan(\alpha) \cdot \tan(\beta))$ | |
| $\sin(\alpha + \beta) \leftrightarrow \sin(\alpha) \cdot \cos(\beta) + \cos(\alpha) \cdot \sin(\beta)$ | |
| $\sin(\alpha - \beta) \leftrightarrow \sin(\alpha) \cdot \cos(\beta) - \cos(\alpha) \cdot \sin(\beta)$ | |
| $\cos(\alpha + \beta) \leftrightarrow \cos(\alpha) \cdot \cos(\beta) - \sin(\alpha) \cdot \sin(\beta)$ | |
| $\cos(\alpha - \beta) \leftrightarrow \cos(\alpha) \cdot \cos(\beta) + \sin(\alpha) \cdot \sin(\beta)$ | |
| $\tan(2 \cdot \alpha) \leftrightarrow 2 \cdot \tan(\alpha)/(1 - \tan(\alpha)^2)$ | |
| $\tan(-1 \cdot \alpha) \leftrightarrow -1 \cdot \tan(\alpha)$ | $\cos(-1 \cdot \alpha) \leftrightarrow \cos(\alpha)$ |
| $\tan(\alpha) \leftrightarrow \sin(\alpha)/\cos(\alpha)$ | $\cos(\pi - \alpha) \leftrightarrow -1 \cdot \cos(\alpha)$ |
| $\tan(\alpha/2) \leftrightarrow \sin(\alpha)/(1 + \cos(\alpha))$ | $\sin(\pi + \alpha) \leftrightarrow -1 \cdot \sin(\alpha)$ |
| $\tan(\pi + \alpha) \leftrightarrow \tan(\alpha)$ | $\sin(\pi - \alpha) \leftrightarrow \sin(\alpha)$ |
| $\tan(\pi - \alpha) \leftrightarrow -1 \cdot \tan(\alpha)$ | $\cos((\pi/2) - \alpha) \leftrightarrow \sin(\alpha)$ |
| $\sin(\alpha)^2 + \cos(\alpha)^2 \leftrightarrow 1$ | $\cos(2 \cdot \alpha) \leftrightarrow \cos(\alpha)^2 - \sin(\alpha)^2$ |
| $\sin(-1 \cdot \alpha) \leftrightarrow -1 \cdot \sin(\alpha)$ | $\cos(2 \cdot \alpha) \leftrightarrow 2 \cdot \cos(\alpha)^2 - 1$ |
| $\sin((\pi/2) - \alpha) \leftrightarrow \cos(\alpha)$ | $\cos(2 \cdot \alpha) \leftrightarrow 1 - 2 \cdot \sin(\alpha)^2$ |
| $\sin(2 \cdot \alpha) \leftrightarrow 2 \cdot \sin(\alpha) \cdot \cos(\alpha)$ | $\cos(\pi + \alpha) \leftrightarrow -1 \cdot \cos(\alpha)$ |

levels and hence there are more positions where axioms could be applied. Furthermore, some axioms require their root node to be several nodes above the leafs of the equation tree because of the depth of the subtree they modify. An example for this is $(\tan(\alpha) - \tan(\beta))/(1 + \tan(\alpha) \cdot \tan(\beta)) \rightarrow \tan(\alpha - \beta)$ which enforces the root node to be at most three levels above the leafs.

For our robustness experiments, we additionally created a dataset whose equations require more than five steps to transform the left side into the right side. Its distribution of depth and level of transformation root is given in table 6. It reveals similar patterns to the test dataset of our main experiments, although it shifts the data towards deeper trees. This can be explained by the fact that deeper trees allow for more complex equations which consequently allow to derive equal expressions in more steps than the shallow trees.

For our second robustness experiment, we create a dataset that consists of deeper trees than seen during training. Its distribution is shown in table 7. Since there are are enough possible expressions for each

Table 5: Count of Samples by Depth of Tree and Level of Transformation Root

| Depth of Tree | Level of Transformation Root | | | | | | Sum |
|---|---|---|---|---|---|---|---|
| | 1 | 2 | 3 | 4 | 5 | 6 | |
| 3 | 0.65% | 1.51% | 0.00% | 0.00% | 0.00% | 0.00% | 2.16% |
| 4 | 2.17% | 3.20% | 3.91% | 0.00% | 0.00% | 0.00% | 9.28% |
| 5 | 3.47% | 4.69% | 5.59% | 6.46% | 0.00% | 0.00% | 20.22% |
| 6 | 4.43% | 5.31% | 7.68% | 6.33% | 7.39% | 0.00% | 31.13% |
| 7 | 4.31% | 4.81% | 7.42% | 8.17% | 6.19% | 6.30% | 37.21% |
| Sum | 15.03% | 19.51% | 24.60% | 20.97% | 13.58% | 6.30% | |

Table 6: Count of Samples by Depth of Tree and Level of Transformation Root for Nine Steps

| Depth of Tree | Level of Transformation Root | | | | | | Sum |
|---|---|---|---|---|---|---|---|
| | 1 | 2 | 3 | 4 | 5 | 6 | |
| 3 | 0.22% | 0.33% | 0.00% | 0.00% | 0.00% | 0.00% | 0.55% |
| 4 | 0.84% | 1.65% | 1.59% | 0.00% | 0.00% | 0.00% | 4.08% |
| 5 | 2.12% | 3.39% | 4.49% | 3.99% | 0.00% | 0.00% | 13.99% |
| 6 | 3.64% | 5.62% | 8.66% | 8.20% | 6.46% | 0.00% | 32.56% |
| 7 | 4.21% | 6.04% | 8.79% | 11.39% | 10.29% | 8.11% | 48.82% |
| Sum | 11.02% | 17.03% | 23.53% | 23.57% | 16.74% | 8.11% | |

considered depth, we decided to keep the dataset balanced with regard to the depth. Considering the level of the root node, the dataset shows a similar trend as the dataset for the main experiments before which can be explained by the same underlying patterns.

Figure 10 visualizes the distribution of the axioms in our dataset. It is clearly visible that only a small fraction of about 25 axioms appear in more than 1.5% of the samples while the vast majority appears much less frequent, with 35 axioms appearing in less than 0.25% of all samples. While creating the dataset, we balance the probabilities for selecting axioms as described in section 3. Yet, it is standing to reason that in the generated data the axioms are not equally distributed because the number of expressions where each axiom can be applied differs largely between the axioms.

The most common axioms we identify in our dataset are simple identities which only depend on a single variable and few functions or operators, for example $\alpha + 0 \rightarrow \alpha$, $\alpha^1 \rightarrow \alpha$, $\alpha \cdot 1 \rightarrow \alpha$, or $e^{\ln(\alpha)} \rightarrow \alpha$. On the other hand, there is a wide range of axioms being used much more seldom. Those axioms generally depend on more variables, functions, and operators or do not have any variables at all. Examples include $\tan(\alpha + \beta) \rightarrow (\tan(\alpha) + \tan(\beta))/(1 - \tan(\alpha) \cdot \tan(\beta))$, $\cos(\alpha - \beta) \rightarrow \cos(\alpha) \cdot \cos(\beta) + \sin(\alpha) \cdot \sin(\beta)$, $2 + 1 \rightarrow 3$, or $3 \rightarrow (-1) \cdot 3$. It is easy to see that, compared to the axioms above, those axioms can only match a much smaller subset of possible expressions. Consequently, there is a lower chance of sampling them during the data generation process since the generator can only sample one of the matching axioms.

However, the influence of the frequency of an axiom on its prediction accuracy is negligible. Using the predictions obtained from TreePointerNet, we only found a very weak correlation of $\rho = 0.073$ between the fraction of correctly identified instances of an axiom and its share in the dataset. Thus, in line with section 5.4, there are other factors influencing the chance of correctly identifying an axiom like the uniqueness of its tree structure.

Table 7: Count of Samples by Depth of Tree and Level of Transformation Root on Deep Trees

| Depth of Tree | Level of Transformation Root | | | | | | | | | | | Sum |
|---|---|---|---|---|---|---|---|---|---|---|---|---|
| | 1 | 2 | 3 | 4 | 5 | 6 | 7 | 8 | 9 | 10 | 11 | |
| 8 | 2.11% | 2.11% | 3.27% | 3.27% | 4.26% | 2.70% | 3.27% | 0.00% | 0.00% | 0.00% | 0.00% | 20.99% |
| 9 | 1.67% | 1.98% | 2.61% | 3.27% | 3.14% | 4.17% | 2.39% | 2.17% | 0.00% | 0.00% | 0.00% | 21.41% |
| 10 | 1.32% | 1.71% | 2.15% | 2.52% | 2.46% | 2.77% | 2.66% | 1.67% | 1.87% | 0.00% | 0.00% | 19.12% |
| 11 | 1.51% | 1.19% | 1.80% | 2.06% | 2.72% | 2.55% | 2.79% | 2.92% | 1.62% | 1.43% | 0.00% | 20.59% |
| 12 | 0.83% | 0.68% | 1.56% | 1.93% | 2.06% | 2.06% | 2.22% | 1.82% | 2.35% | 1.30% | 1.08% | 17.89% |
| Sum | 7.44% | 7.66% | 11.39% | 13.06% | 14.64% | 14.25% | 13.33% | 8.58% | 5.84% | 2.72% | 1.08% | |

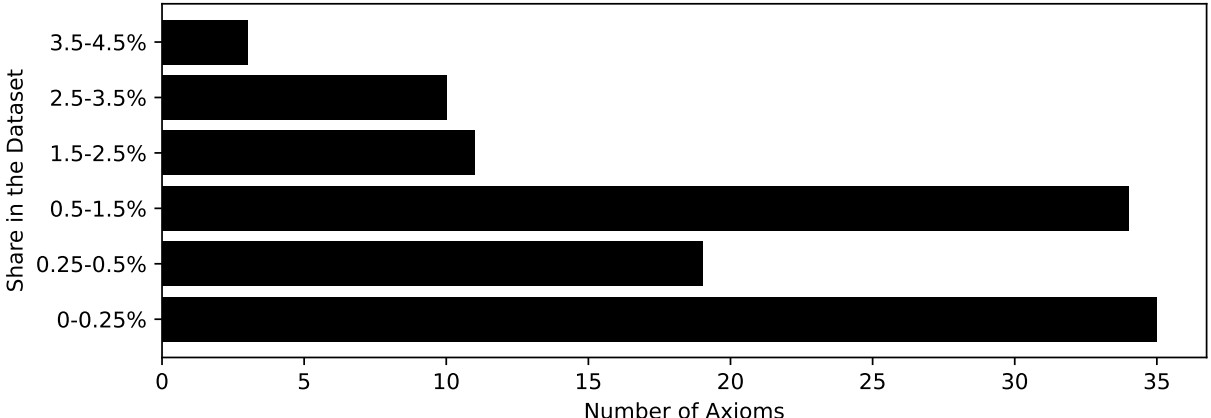

Figure 10: Distribution of Axioms in the Dataset

## C  Hyperparameters

This section lists all relevant hyperparameters we used for training our models and the resulting number of trainable parameters. The hyperparameters were found using Optuna (Akiba et al., 2019) with the TPE sampler.

We use the identical search space for all models. For the size of embeddings, we search powers of two between 32 and 512. For the number of encoder and decoder layers we search values between 1 and 8. For the hidden size of the encoder and decoder layers, we search powers of two between 32 and 1024. For the transformer model, we search the number of attention heads for all power of two between 4 and 16. In TreePointerNet and SeqPointer, we search the number of pointer heads between 1 and 3. For the LSTM, we check if we should use a unidirectional or a bidirectional encoder. Also, we check if attention is beneficial. Finally, we optimize the ordering of the target sequence to be either `axiom position` or `position axiom`.

In table table 8 we list the values of all transformer-based models. The LSTM networks consists of a one-layer unidirectional LSTM encoder and decoder with an embedding size of 512 and a hidden size of 1024. It has 6,496,256 trainable parameters.

## D  Detailed Results for Main Experiments

In this section, we show the detailed results as discussed in section 5. Tables 9 to 12 show the average accuracy ($\pm$ standard deviation over five model runs, each time initialized with a different random seed), i.e. the fraction of correctly transformed equations grouped by the number of required transformations and the macro average over all groups. The best results are indicated in bold.

Table 8: Hyperparameters Used for our Models

|  |  | TreePointerNet | SeqPointer | Transformer |
|---|---|---|---|---|
| Encoder | Layers | 6 | 5 | 5 |
|  | Attention Heads | 4 | 4 | 4 |
|  | Hidden Size | 1024 | 512 | 512 |
|  | Embedding Size | 128 | 512 | 512 |
| Decoder | Layers | 4 | 5 | 5 |
|  | Attention Heads | 4 | 16 | 16 |
|  | Heads for Pointer | 3 | 4 | - |
|  | Hidden Size | 64 | 1024 | 1024 |
|  | Embedding Size | 64 | 512 | 512 |
| Parameters |  | 2,649,901 | 22,768,644 | 22,768,644 |

Table 9: Results for all Models on Equations of up to 5 Required Transformation Steps

| Model | Number of Transformation Steps | | | | | Average |
|---|---|---|---|---|---|---|
|  | 1 | 2 | 3 | 4 | 5 |  |
| TreePointerNet (ours) | $\mathbf{88.43\%}_{\pm0.09\%}$ | $\mathbf{82.02\%}_{\pm0.51\%}$ | $\mathbf{74.43\%}_{\pm0.36\%}$ | $\mathbf{66.53\%}_{\pm0.84\%}$ | $\mathbf{59.05\%}_{\pm0.80\%}$ | $\mathbf{74.09\%}$ |
| SeqPointer | $88.41\%_{\pm0.06\%}$ | $70.83\%_{\pm0.67\%}$ | $55.84\%_{\pm0.64\%}$ | $45.52\%_{\pm0.66\%}$ | $36.74\%_{\pm0.95\%}$ | $59.47\%$ |
| Transformer | $88.43\%_{\pm0.24\%}$ | $70.27\%_{\pm0.94\%}$ | $55.76\%_{\pm1.65\%}$ | $45.29\%_{\pm1.64\%}$ | $36.65\%_{\pm0.72\%}$ | $59.28\%$ |
| LSTM | $85.49\%_{\pm0.19\%}$ | $70.57\%_{\pm0.39\%}$ | $57.29\%_{\pm0.41\%}$ | $46.61\%_{\pm0.25\%}$ | $37.40\%_{\pm1.12\%}$ | $59.47\%$ |

Table 9 present the main results. The results for robustness experiments are presented in table 10 for the experiments on deeper trees and table 11 for the experiments on equations requiring up to 9 transformation steps. The results of the ablation study on TreePointerNet are given in table 12.

# E    Representing the Index as a Separate Token

As described in section 4.1, we experimented with a second tokenization strategy where the index is represented as a separate token from the operator, constant, function, or variable. We evaluated our SeqPointer, transformer, and LSTM baselines with this alternative strategy on the same data and experiments conducted in section 5. The results are presented in table 13 (main experiment), table 14 (deep input trees), and table 15 (more transformation steps). In all tables we include TreePointerNet for reference, although it is not affected by the alternative tokenization strategy.

With regard to the main experiment, it can be observed that the effect of the tokenization on the performance of the models is small. While the performance of the transformer model increases by 0.66 percentage points, the SeqPointer looses 1.32 percentage points. Similarly, the LSTM looses 2.87 percentage points.

Considering the equations requiring up to 9 transformation steps, the same pattern can be observed. Here, the transformer gains 1.35 percentage points while the SeqPointer and the LSTM both loose 1.56 and 3.06 percentage points.

Slightly different results can be observed when considering the experiments on deeper trees. Here, the separate index token seems to be more beneficial. Both the transformer as well as the SeqPointer marginally gain accuracy, namely 3.99 and 1.2 percentage points. However, the LSTM looses again, namely 2.34 percentage points.

Summarizing, one can observe that this tokenization strategy proved to be beneficial for the transformer whose performance increases in all experiments. On the other hand, the LSTM looses accuracy on all experiments. The SeqPointer yielded mixed results, it could only benefit when considering the deep input trees. Most importantly, in all cases the performance of the sequential models distinctly stays below the

Table 10: Robustness Study for all Models on Equations of Deeper Parse Trees

| Model | Number of Transformation Steps | | | | | Average |
|---|---|---|---|---|---|---|
| | 1 | 2 | 3 | 4 | 5 | |
| TreePointerNet (ours) | $\mathbf{58.48\%}_{\pm 0.77\%}$ | $\mathbf{49.52\%}_{\pm 1.23\%}$ | $\mathbf{40.58\%}_{\pm 1.37\%}$ | $\mathbf{34.13\%}_{\pm 1.70\%}$ | $\mathbf{29.71\%}_{\pm 1.09\%}$ | $\mathbf{42.48\%}$ |
| SeqPointer | $50.49\%_{\pm 1.48\%}$ | $36.34\%_{\pm 0.74\%}$ | $26.31\%_{\pm 0.30\%}$ | $19.80\%_{\pm 0.70\%}$ | $15.21\%_{\pm 0.38\%}$ | $29.63\%$ |
| Transformer | $49.97\%_{\pm 0.70\%}$ | $34.09\%_{\pm 0.88\%}$ | $24.32\%_{\pm 1.00\%}$ | $18.73\%_{\pm 0.73\%}$ | $13.43\%_{\pm 0.81\%}$ | $28.11\%$ |
| LSTM | $50.12\%_{\pm 0.85\%}$ | $31.77\%_{\pm 0.52\%}$ | $23.11\%_{\pm 1.39\%}$ | $17.92\%_{\pm 0.58\%}$ | $12.97\%_{\pm 0.79\%}$ | $27.18\%$ |

Table 11: Robustness Study for all Models on Equations of up to 9 Required Transformation Steps

| Model | Number of Transformation Steps | | | | Average |
|---|---|---|---|---|---|
| | 6 | 7 | 8 | 9 | |
| TreePointerNet (ours) | $\mathbf{58.49\%}_{\pm 0.85\%}$ | $\mathbf{54.63\%}_{\pm 1.43\%}$ | $\mathbf{51.34\%}_{\pm 1.74\%}$ | $\mathbf{49.14\%}_{\pm 1.81\%}$ | $\mathbf{53.40\%}$ |
| SeqPointer | $34.47\%_{\pm 0.80\%}$ | $32.78\%_{\pm 0.47\%}$ | $30.15\%_{\pm 0.56\%}$ | $27.95\%_{\pm 0.86\%}$ | $31.34\%$ |
| Transformer | $33.16\%_{\pm 1.05\%}$ | $32.80\%_{\pm 0.88\%}$ | $29.42\%_{\pm 1.03\%}$ | $27.64\%_{\pm 0.61\%}$ | $30.75\%$ |
| LSTM | $36.18\%_{\pm 0.88\%}$ | $32.26\%_{\pm 0.67\%}$ | $29.95\%_{\pm 0.40\%}$ | $28.74\%_{\pm 0.69\%}$ | $31.78\%$ |

accuracy of TreePointerNet and therefore does not qualitatively change the results we obtained for our experiments in section 5.

## F  Exploring Large Language Models for the SETI Task

Listing 1: Prompt Template for LLMs

```
Give a step towards transforming an equation into an equivalent form. Give me the
    axiom to apply and indicate its exact position, where to apply it.
Example: Input: ( ( sin ( cos ( 0 ) ) * cos ( sin ( 0 ) ) ) - ( cos ( cos ( 0 ) ) *
    sin ( sin ( 0 ) ) ) ) = sin_0 ( ( cos_0 ( 0_0 ) -_0 sin_1 ( 0_1 ) ) ) , Answer: (
    sin(x)*cos(y))-(cos(x)*sin(y))<-sin((x-y)), Position: sin_0
Example: Input: ( sin ( tan ( ( 1 / 2 ) ) ) / cos ( tan ( ( 1 / 2 ) ) ) ) = ( sin_0
    ( tan_0 ( ( 1_0 /_1 2_0 ) ) ) /_0 cos_0 ( tan_1 ( ( ( x_0 **_0 0_0 ) /_2 2_1 ) )
    ) ) , Answer: 1<-x**0, Position: **_0
Example: Input: ( ( sin ( ( ( 1 - 1 ) + ( 1 ** 1 ) ) ) / 1 ) ** ( -1 ) ) = ( ( sin_0
    ( ( ( 1_0 -_0 1_1 ) +_0 1_3 ) ) /_0 1_2 ) **_0 ( -1_0 ) ) , Answer: x**1<-x,
    Position: 1_3
Input: cos ( ( 2 * ( ( -1 ) * sin ( 2 ) ) ) ) = ( ( 2_0 *_0 ( cos_0 ( ( ( -1_0 ) *_1
    sin_0 ( 2_2 ) ) ) **_0 2_1 ) ) -_0 1_0 ) , Answer:
```

In this work, we develop TreePointerNet as a parameter-efficient neural network architectures for the SETI task and compare it to various neural baselines. Both TreePointerNet and the baseline models are trained to explicitly solve the SETI task. However, in recent years, large language models (LLMs) emerged as multi-purpose neural networks that were designed to receive a prompt stating a problem in natural language and then output a solution. In this section, we briefly want to explore the capabilities of various LLMs for the SETI task.

As a first experiment, we used o1 (OpenAI, 2024) with a prompt like shown in listing 1. The reasoning time of o1 was between 20 and 50 seconds per transformation step. Thus, due to the long reasoning time and the limited access to this model, we could not evaluate it for all equations in our dataset. Instead, we tested it on two equations requiring only one transformation step and two further equations requiring more transformation steps. All of them were solved correctly by TreePointerNet.

We found that o1 was able to yield an output in the correct format, i.e. a pair of an axiom and a position, in all tested cases. Moreover, it was able to solve all examples which require only one transformation step

Table 12: Ablation Study for our TreePointerNet

| Model | Number of Transformation Steps | | | | | Average |
|---|---|---|---|---|---|---|
| | 1 | 2 | 3 | 4 | 5 | |
| TreePointerNet (ours) | **88.43**%$_{\pm 0.09\%}$ | **82.02**%$_{\pm 0.51\%}$ | **74.43**%$_{\pm 0.36\%}$ | **66.53**%$_{\pm 0.84\%}$ | **59.05**%$_{\pm 0.80\%}$ | **74.09**% |
| w/o ME | 87.82%$_{\pm 0.34\%}$ | 69.01%$_{\pm 0.28\%}$ | 52.59%$_{\pm 0.83\%}$ | 42.84%$_{\pm 0.52\%}$ | 34.31%$_{\pm 0.50\%}$ | 57.31% |
| w/o pointer | 6.76%$_{\pm 0.35\%}$ | 0.60%$_{\pm 0.08\%}$ | 0.16%$_{\pm 0.05\%}$ | 0.07%$_{\pm 0.04\%}$ | 0.05%$_{\pm 0.03\%}$ | 1.53% |
| w/o pointer + ME | 87.76%$_{\pm 0.19\%}$ | 68.87%$_{\pm 0.42\%}$ | 52.29%$_{\pm 0.79\%}$ | 42.67%$_{\pm 0.51\%}$ | 34.10%$_{\pm 0.42\%}$ | 57.14% |

Table 13: Results for all Models on Equations of up to 5 Required Transformation Steps with Separate Index Token

| Model | Number of Transformation Steps | | | | | Average |
|---|---|---|---|---|---|---|
| | 1 | 2 | 3 | 4 | 5 | |
| TreePointerNet (ours) | 88.43%$_{\pm 0.09\%}$ | **82.02**%$_{\pm 0.51\%}$ | **74.43**%$_{\pm 0.36\%}$ | **66.53**%$_{\pm 0.84\%}$ | **59.05**%$_{\pm 0.80\%}$ | **74.09**% |
| SeqPointer | **88.58**%$_{\pm 0.09\%}$ | 69.32%$_{\pm 0.98\%}$ | 53.95%$_{\pm 0.88\%}$ | 44.05%$_{\pm 1.10\%}$ | 34.83%$_{\pm 0.91\%}$ | 58.15% |
| Transformer | 88.57%$_{\pm 0.08\%}$ | 71.47%$_{\pm 0.72\%}$ | 56.13%$_{\pm 0.84\%}$ | 45.94%$_{\pm 1.14\%}$ | 37.60%$_{\pm 0.63\%}$ | 59.94% |
| LSTM | 84.18%$_{\pm 0.51\%}$ | 67.81%$_{\pm 1.00\%}$ | 54.33%$_{\pm 0.72\%}$ | 42.67%$_{\pm 0.98\%}$ | 34.00%$_{\pm 0.92\%}$ | 56.60% |

correctly but failed on the equations which require more transformation steps. However, since o1 is a closed-source model that can change at any time, the acquired results are not representative and can only be considered as a rough test towards the capabilities of LLMs.

In addition, we tested two open-source LLMs that were trained for mathematical reasoning tasks and can be run locally, namely Llemma (Azerbayev et al., 2024) and Qwen-2.5 Math (Yang et al., 2024). However, using above prompt none of these models were able to solve our task on various tested equations. Instead of an axiom and position for one step, they output a textual deduction of the equivalence.

To make the task easier, we experimented with chain-of-thought prompting since it proofed beneficial for reasoning tasks in other settings (Wei et al., 2022). Precisely, we added textual descriptions to the input examples like `Example: Input: ( ( sin ( cos ( 0 ) ) * cos ( sin ( 0 ) ) ) - ( cos ( cos ( 0 ) ) * sin ( sin ( 0 ) ) ) ) = sin_0 ( ( cos_0 ( 0_0 ) -_0 sin_1 ( 0_1 ) ) ) . Answer: The first sin can be transformed using the axiom (sin(x)*cos(y))-(cos(x)*sin(y))<-sin((x-y)). The answer is (sin(x)*cos(y))-(cos(x)*sin(y))<-sin((x-y)) sin_0.` There was no change in the models' output. Finally, we removed the indices from the input expression and let the model deduce the position. However, the models still only yields a textual description of the equivalence.

We conclude that the tested open-source LLMs are not able to solve our task with simple prompting techniques. Further research is necessary to integrate the SETI task into LLMs, be it by finetuning them on the task or developing more sophisticated prompting techniques.

## G  Examples

For illustration, we present a few equations solved by our TreePointerNet in this section. Tables 16 to 19 show examples of equations that the model can derive in $n = 5$ steps. In tables 20 and 21, we show examples on our out-of-distribution tests on equations that can be derived in $n = 9$ steps.

In each table, the first $n$ rows correspond to a step of the iterative transformation process and the last row shows the final result. For every step $i \leq n$, the first column shows the input equation given to the network. The second and third columns show the axiom and the position as they are predicted by the network. The subsequent row $i+1$ presents the equation after applying the predicted axiom at the predicted position from step $i$.

Table 14: Robustness Study for all Models on Equations of Deeper Parse Trees with Separate Index Token

| Model | Number of Transformation Steps | | | | | Average |
|-------|------|------|------|------|------|---------|
| | 1 | 2 | 3 | 4 | 5 | |
| TreePointerNet (ours) | **58.48%**$_{\pm 0.77\%}$ | **49.52%**$_{\pm 1.23\%}$ | **40.58%**$_{\pm 1.37\%}$ | **34.13%**$_{\pm 1.70\%}$ | **29.71%**$_{\pm 1.09\%}$ | **42.48%** |
| SeqPointer | 55.82%$_{\pm 0.53\%}$ | 39.16%$_{\pm 0.79\%}$ | 26.81%$_{\pm 1.67\%}$ | 20.21%$_{\pm 1.19\%}$ | 14.25%$_{\pm 0.32\%}$ | 31.25% |
| Transformer | 54.82%$_{\pm 0.74\%}$ | 39.52%$_{\pm 1.47\%}$ | 28.22%$_{\pm 1.14\%}$ | 21.75%$_{\pm 1.11\%}$ | 16.20%$_{\pm 0.16\%}$ | 32.10% |
| LSTM | 47.50%$_{\pm 0.42\%}$ | 29.37%$_{\pm 1.62\%}$ | 20.30%$_{\pm 0.73\%}$ | 15.90%$_{\pm 1.01\%}$ | 11.14%$_{\pm 0.58\%}$ | 24.84% |

Table 15: Robustness Study for all Models on Equations of up to 9 Required Transformation Steps with Separate Index Token

| Model | Number of Transformation Steps | | | | Average |
|-------|------|------|------|------|---------|
| | 6 | 7 | 8 | 9 | |
| TreePointerNet (ours) | **58.49%**$_{\pm 0.85\%}$ | **54.63%**$_{\pm 1.43\%}$ | **51.34%**$_{\pm 1.74\%}$ | **49.14%**$_{\pm 1.81\%}$ | **53.40%** |
| SeqPointer | 32.58%$_{\pm 0.67\%}$ | 31.44%$_{\pm 0.25\%}$ | 28.64%$_{\pm 0.78\%}$ | 26.46%$_{\pm 1.18\%}$ | 29.78% |
| Transformer | 35.25%$_{\pm 0.88\%}$ | 33.68%$_{\pm 0.69\%}$ | 30.56%$_{\pm 0.76\%}$ | 28.57%$_{\pm 1.03\%}$ | 32.01% |
| LSTM | 33.16%$_{\pm 1.28\%}$ | 29.83%$_{\pm 1.16\%}$ | 26.78%$_{\pm 1.11\%}$ | 25.12%$_{\pm 1.26\%}$ | 28.72% |

In each equation, we use the symbol $\wedge$ to represent the power instead of the commonly used superscript. For example, we write $x \wedge 2$ instead of $x^2$. By doing so, we are able to indicate the position of the power as predicted by the network in the same way we do it for the other binary operators (e.g. $+_0$, $\wedge_1$).

Since a new expression tree is generated in each transformation step, the indices of the mathematical tokens are reinitialized in each row of the tables. As a consequence, at any step a predicted position might refer to another instance of the token as it did in the step before. For example, in table 16, the rightmost addition summing the constants 2 and 1 is denoted as $+_1$ in step 1 because another addition exists in the same equation. However, in step 2 it becomes $+_0$ since the other addition is eliminated by the axiom chosen in step 1.

Table 16: Transformations for $e^{\ln(1)+\ln(2+1)} = 3$

| Step | Equation | Predicted Axiom | Predicted Position |
|------|----------|-----------------|--------------------|
| 1 | $e_0 \wedge_0 (\ln_0(1_0) +_0 \ln_1(2_0 +_1 1_1)) = 3$ | $\ln(\alpha) + \ln(\beta) \to \ln(\alpha \cdot \beta)$ | $+_0$ |
| 2 | $e_0 \wedge_0 \ln_0(1_0 \cdot_0 (2_0 +_0 1_1)) = 3$ | $2+1 \to 3$ | $+_0$ |
| 3 | $e_0 \wedge_0 \ln_0(1_0 \cdot_0 3_0) = 3$ | $\alpha \cdot \beta \to \beta \cdot \alpha$ | $\cdot_0$ |
| 4 | $e_0 \wedge_0 \ln_0(3_0 \cdot_0 1_0) = 3$ | $\alpha \cdot 1 \to \alpha$ | $\cdot_0$ |
| 5 | $e_0 \wedge_0 \ln_0(3_0) = 3$ | $e^{\ln(\alpha)} \to \alpha$ | $\wedge_0$ |
| Final | $3_0 = 3$ | | |

Table 17: Transformations for $\tan(((\ln(e)+z)\cdot(x^0))^{x^0}) = \tan(1+z)$

| Step | Equation | Predicted Axiom | Predicted Position |
|------|----------|-----------------|--------------------|
| 1 | $\tan_0(((\ln_0(e_0) +_0 z_0) \cdot_0 (x_0 \wedge_1 0_0)) \wedge_0 (x_1 \wedge_2 0_1)) = \tan(1+z)$ | $\alpha^0 \to 1$ | $\wedge_2$ |
| 2 | $\tan_0(((\ln_0(e_0) +_0 z_0) \cdot_0 (x_0 \wedge_1 0_0)) \wedge_0 1_0) = \tan(1+z)$ | $\alpha^1 \to \alpha$ | $\wedge_0$ |
| 3 | $\tan_0((\ln_0(e_0) +_0 z_0) \cdot_0 (x_0 \wedge_0 0_0)) = \tan(1+z)$ | $\alpha^0 \to 1$ | $\wedge_0$ |
| 4 | $\tan_0((\ln_0(e_0) +_0 z_0) \cdot_0 1_0) = \tan(1+z)$ | $x \cdot 1 \to x$ | $\cdot_0$ |
| 5 | $\tan_0(\ln_0(e_0) +_0 z_0) = \tan(1+z)$ | $\ln(e) \to 1$ | $\ln_0$ |
| Final | $\tan_0(1_0 +_0 z_0) = \tan(1+z)$ | | |

Table 18: Transformations for $\sin(\pi/2 - \sin((-1)\cdot z \cdot 2)) = \cos(2 \cdot \sin(z) \cdot \cos(z))$

| Step | Equation | Predicted Axiom | Predicted Position |
|------|----------|-----------------|--------------------|
| 1 | $\sin_0((\pi_0/_0 2_0) -_0 \sin_1((-1_0) \cdot_0 (z_0 \cdot_1 2_1))) = \cos((2 \cdot \sin(z)) \cdot \cos(z))$ | $\sin(-1 \cdot \alpha) \to (-1) \cdot \sin(\alpha)$ | $\sin_1$ |
| 2 | $\sin_0((\pi_0/_0 2_0) -_0 ((-1_0) \cdot_0 \sin_1(z_0 \cdot_1 2_1))) = \cos((2 \cdot \sin(z)) \cdot \cos(z))$ | $\sin(\pi/2 - \alpha) \to \cos(\alpha)$ | $\sin_0$ |
| 3 | $\cos_0((-1_0) \cdot_0 \sin_0(z_0 \cdot_1 2_0)) = \cos((2 \cdot \sin(z)) \cdot \cos(z))$ | $\cos(-1 \cdot \alpha) \to \cos(\alpha)$ | $\cos_0$ |
| 4 | $\cos_0(\sin_0(z_0 \cdot_0 2_0)) = \cos((2 \cdot \sin(z)) \cdot \cos(z))$ | $\alpha \cdot \beta \to \beta \cdot \alpha$ | $\cdot_0$ |
| 5 | $\cos_0(\sin_0(2_0 \cdot_0 z_0)) = \cos((2 \cdot \sin(z)) \cdot \cos(z))$ | $\sin(2 \cdot \alpha) \to 2 \cdot \sin(\alpha) \cdot \cos(\alpha)$ | $\sin_0$ |
| Final | $\cos_0((2_0 \cdot_1 \sin_0(z_1)) \cdot_0 \cos_1(z_0)) = \cos((2 \cdot \sin(z)) \cdot \cos(z))$ | | |

Table 19: Transformations for $2 \cdot \cos(1^{-1} \cdot x)^2 - (\sin(x)^2 + \cos(x)^2) = \cos(2 \cdot x)$

| Step | Equation | Predicted Axiom | Predicted Position |
|------|----------|-----------------|--------------------|
| 1 | $(2_0 \cdot_0 (\cos_0((1_0 \wedge_3 (-1_0)) \cdot_1 x_2) \wedge_0 2_1)) -_0 ((\sin_0(x_0) \wedge_1 2_2) +_0 (\cos_1(x_1) \wedge_2 2_3)) = \cos(2 \cdot x)$ | $1^{-1} \to 1$ | $\wedge_3$ |
| 2 | $(2_2 \cdot_0 (\cos_1(1_0 \cdot_1 x_2) \wedge_2 2_3)) -_0 ((\sin_0(x_0) \wedge_0 2_0) +_0 (\cos_0(x_1) \wedge_1 2_1)) = \cos(2 \cdot x)$ | $\sin(\alpha)^2 + \cos(\alpha)^2 \to 1$ | $+_0$ |
| 3 | $(2_0 \cdot_0 (\cos_0((1_0 \cdot_1 x_0)) \wedge_0 2_1)) -_0 1_1 = \cos(2 \cdot x)$ | $2 \cdot (\cos(\alpha)^2) - 1 \to \cos(2 \cdot \alpha)$ | $-_0$ |
| 4 | $\cos_0(2_0 \cdot_0 (1_0 \cdot_1 x_0)) = \cos(2 \cdot x)$ | $\alpha \cdot (\beta \cdot \gamma) \to (\alpha \cdot \beta) \cdot \gamma$ | $\cdot_0$ |
| 5 | $\cos_0((2_0 \cdot_1 1_0) \cdot_0 x_0) = \cos(2 \cdot x)$ | $\alpha \cdot 1 \to \alpha$ | $\cdot_1$ |
| Final | $\cos_0(2_0 \cdot_0 x_0) = \cos(2 \cdot x)$ | | |

Table 20: Transformations for $(\sin((\pi - ((-1) \cdot (-1)))) \cdot 1) \cdot (1 + \cos(x^{x^{\ln(1)}})^{(-1)^1}) = \sin(1)/(1+\cos(1))$

| Step | Equation | Predicted Axiom | Predicted Position |
|------|----------|-----------------|--------------------|
| 1 | $(\sin_0(\pi_0 -_0 ((-1_0) \cdot_2 (-1_1))) \cdot_1 1_0) \cdot_0 ((1_1 +_0 \cos_0((x_0 \wedge_2 x_1) \wedge_1 \ln_0(1_2))) \wedge_0 ((-1_2) \wedge_3 1_3)) = \sin(1)/(1+\cos(1))$ | $\sin(\pi - \alpha) \to \sin(\alpha)$ | $\sin_0$ |
| 2 | $(\sin_0((-1_1) \cdot_2 (-1_2)) \cdot_1 1_3) \cdot_0 ((1_0 +_0 \cos_0((x_0 \wedge_2 x_1) \wedge_1 \ln_0(1_1))) \wedge_0 ((-1_0) \wedge_3 1_2)) = \sin(1)/(1+\cos(1))$ | $\ln(1) \to 0$ | $\ln_0$ |
| 3 | $(\sin_0((-1_0) \cdot_2 (-1_1)) \cdot_1 1_0) \cdot_0 ((1_1 +_0 \cos_0((x_0 \wedge_2 x_1) \wedge_1 0_0)) \wedge_0 ((-1_2) \wedge_3 1_2)) = \sin(1)/(1+\cos(1))$ | $(\alpha^\beta)^\gamma \to \alpha^{\beta \cdot \gamma}$ | $\wedge_1$ |
| 4 | $(\sin_0((-1_0) \cdot_2 (-1_1)) \cdot_1 1_0) \cdot_0 ((1_1 +_0 \cos_0(x_1 \wedge_2 (x_0 \cdot_3 0_0))) \wedge_0 ((-1_2) \wedge_1 1_2)) = \sin(1)/(1+\cos(1))$ | $\alpha \cdot 0 \to 0$ | $\cdot_3$ |
| 5 | $(\sin_0((-1_0) \cdot_2 (-1_1)) \cdot_1 1_0) \cdot_0 ((1_1 +_0 \cos_0(x_0 \wedge_1 0_0)) \wedge_0 ((-1_2) \wedge_2 1_2)) = \sin(1)/(1+\cos(1))$ | $(-1) \cdot (-1) \to 1$ | $\cdot_2$ |
| 6 | $(\sin_0(1_3) \cdot_1 1_0) \cdot_0 ((1_1 +_0 \cos_0(x_0 \wedge_1 0_0)) \wedge_0 ((-1_0) \wedge_2 1_2)) = \sin(1)/(1+\cos(1))$ | $\alpha^1 \to \alpha$ | $\wedge_2$ |
| 7 | $(\sin_0(1_1) \cdot_1 1_0) \cdot_0 ((1_2 +_0 \cos_0(x_0 \wedge_1 0_0)) \wedge_0 (-1_0)) = \sin(1)/(1+\cos(1))$ | $\alpha \cdot \beta^{-1} \to \alpha/\beta$ | $\cdot_0$ |
| 8 | $(\sin_0(1_1) \cdot_0 1_0)/_0(1_2 +_0 \cos_0(x_0 \wedge_0 0_0)) = \sin(1)/(1+\cos(1))$ | $\alpha^0 \to 1$ | $\wedge_0$ |
| 9 | $(\sin_0(1_2) \cdot_0 1_1)/_0(1_0 +_0 \cos_0(1_3)) = \sin(1)/(1+\cos(1))$ | $\alpha \cdot 1 \to \alpha$ | $\cdot_0$ |
| Final | $\sin_0(1_0)/_0(1_1 +_0 \cos_0(1_2)) = \sin(1)/(1+\cos(1))$ | | |

Table 21: Transformations for $(((\sin(x)^2) \cdot (-1)) + ((-1) \cdot (\cos(x)^2))) \cdot \cos((\pi \cdot 2^{-1}) - (\pi - y)) = (-1) \cdot \sin(y)$

| Step | Equation | Predicted Axiom | Predicted Position |
|---|---|---|---|
| 1 | $(((\sin_0(x_0) \wedge_0 2_0) \cdot_1 (-1_0)) +_0 ((-1_1) \cdot_2 (\cos_0(x_1) \wedge_1 2_1))) \cdot_0 \cos_1((\pi_0 \cdot_3 (2_2 \wedge_2 (-1_2))) -_0 (\pi_1 -_1 y_0)) = (-1) \cdot \sin(y)$ | $\alpha \cdot \beta^{-1} \to \alpha/\beta$ | $\cdot_3$ |
| 2 | $(((\sin_0(x_0) \wedge_0 2_0) \cdot_1 (-1_0)) +_0 ((-1_1) \cdot_2 (\cos_0(x_1) \wedge_1 2_1))) \cdot_0 \cos_1((\pi_0/_0 2_2) -_0 (\pi_1 -_1 y_0)) = (-1) \cdot \sin(y)$ | $\cos((\pi/2) - \alpha) \to \sin(\alpha)$ | $\cos_1$ |
| 3 | $(((\sin_0(x_0) \wedge_0 2_0) \cdot_1 (-1_0)) +_0 ((-1_1) \cdot_2 (\cos_0(x_1) \wedge_1 2_1))) \cdot_0 \sin_1(\pi_0 -_0 y_0) = (-1) \cdot \sin(y)$ | $\alpha + ((-1) \cdot \beta) \to \alpha - \beta$ | $+_0$ |
| 4 | $(((\sin_0(x_0) \wedge_0 2_0) \cdot_1 (-1_0)) -_0 (\cos_0(x_1) \wedge_1 2_1)) \cdot_0 \sin_1(\pi_0 -_1 y_0) = (-1) \cdot \sin(y)$ | $\beta \cdot \alpha \to \alpha \cdot \beta$ | $\cdot_1$ |
| 5 | $(((-1_0) \cdot_1 (\sin_1(x_0) \wedge_0 2_0)) -_1 (\cos_0(x_1) \wedge_1 2_1)) \cdot_0 \sin_0(\pi_0 -_0 y_0) = (-1) \cdot \sin(y)$ | $\alpha - \beta \to \alpha + ((-1) \cdot \beta)$ | $-_1$ |
| 6 | $(((-1_0) \cdot_1 (\sin_0(x_0) \wedge_0 2_0)) +_0 ((-1_1) \cdot_2 (\cos_0(x_1) \wedge_1 2_1))) \cdot_0 \sin_1(\pi_0 -_0 y_0) = (-1) \cdot \sin(y)$ | $(\alpha \cdot \beta) + (\alpha \cdot \gamma) \to \alpha \cdot (\beta + \gamma)$ | $+_0$ |
| 7 | $((-1_0) \cdot_1 ((\sin_0(x_0) \wedge_0 2_0) +_0 (\cos_0(x_1) \wedge_1 2_1))) \cdot_0 \sin_1(\pi_0 -_0 y_0) = (-1) \cdot \sin(y)$ | $\sin(\alpha)^2 + \cos(\alpha)^2 \to 1$ | $+_0$ |
| 8 | $((-1_0) \cdot_1 1_0) \cdot_0 \sin_0(\pi_0 -_0 y_0) = (-1) \cdot \sin(y)$ | $\sin(\pi - \alpha) \to \sin(\alpha)$ | $\sin_0$ |
| 9 | $((-1_0) \cdot_1 1_0) \cdot_0 \sin_0(y_0) = (-1) \cdot \sin(y)$ | $(-1) \cdot 1 \to (-1)$ | $\cdot_1$ |
| Final | $(-1_0) *_0 \sin_0(y_0) = (-1) * \sin(y)$ | | |

