# OpenReview forum: "Identifying Axiomatic Mathematical Transformation Steps using Tree-Structured Pointer Networks"
_TMLR — Accepted by TMLR_

### Review · Reviewer_Dovy · 2024-11-21

**Summary Of Contributions:**

1. Introducing the SETI Task: The authors introduce the new Stepwise Equation Transformation Identification (SETI) task in the field of deep learning for symbolic mathematics. This task focuses on predicting a sequence of axiomatic mathematical transformation steps to transform one mathematical expression into an equivalent one. It fills a gap in the research area by providing a more detailed and step-by-step approach to understanding the equivalence of mathematical expressions.
2. Proposing the TreePointerNet Architecture: A novel neural network architecture, TreePointerNet, is designed. It combines a pointer generator network with a hierarchical-accumulation model for tree-structured input and a unique embedding strategy. This architecture is tailored to solve the SETI task efficiently and effectively, taking advantage of the inherent structure of mathematical expressions.
3. Generating a New Dataset: A new equation dataset is created for the SETI task. The dataset consists of pairs of mathematically equivalent expressions represented as expression trees and includes a wide range of mathematical functions and equations with varying complexity. This dataset is crucial for training and evaluating models on the SETI task and provides a valuable resource for future research in this area.

**Audience:**

Yes

**Broader Impact Concerns:**

1. The proposed axiom transformation method can be used to generate intermediate stepped axiom proving data for mathematical LLMs with significantly low costs.

**Claims And Evidence:**

Yes

**Requested Changes:**

1. The contents are good, but the structure of the paper need to be re-organized.
2. The problem statement might need to be put in an independent section, and improve the clarity of the problem statement in the introduction.
3. The formulation expression of the "ME" operation need to be unified over the whole paper for more clearly presentation. For example, "Our many-to-many embedding layer me then ......".
4. Investivate more recent methods about the axiom proving problem, and represent the experiment results woth more detailed tables and figures.

**Strengths And Weaknesses:**

Strengths:
1. The proposed Stepwise Equation Transformation Identification (SETI) task is novel and significant, opened up a new path in the field of automated mechanical theory proving. The proposed method is not uncessarity relied on LLMs and ceatively combined TreeTransformer and Pointer Network to guarantee the precise appropriation of the formula terms. All of the contributions make this work quite interesting.
2. The theory construction part in section 2 is concrete, with enough formular  representations to clarify the proposed methods.
3. The proposed TreePointerNet has a parameter size of 2.6M, which is far more smaller than the existed methods, which indicates the imporvement of the architecture and training data are effective in the SETI task.
4. The experiment shows the proposed TreePointerNet achieves the best results over other baseline methods on various experiment settings. The ablation of different proposed components successfully exhibites the effectiveness of each of them.

Weaknesses：
1. The major weakness is the language presentation of this paper. The writing style is not quite easy to follow.  Three major types of pitfalls are found: (1) Some of the complex sentences are too long to clearly represent the author's idea. For example, "This approach has the davantage that ...... nodes observed during training" at page 3.  (2) There exists some unnecessary sentenses, some of them  confuse the reader. For example, "We make our code and dataset ...... (after acceptance of the paper)" and "As with all deep learning architectures ...... to an embedding" at page 3 . (3) The excessive use of the linking words cannot individually make the paragraph more logical. For example, in page 2 it includes "In theroy, ... In practice, ... however,  ... Empirically, ... Hence, ... However, ... So far, ... More over , ... Thus, ... Besides ..." in a paragraph with 13 lines.
2. The concept of the proposed task is poorly defined in the Introduction section. Why do not the author put the first paragraph of Section 2, a very clear definition of the task, into the introduction section to clarify the task?
3. The organization of the sections is confusion and do not well paralleled with the introduction. To be psecific, Why other's work about TreeTransformer are described in section 2.1 with so many details? And the dataset constraction part is unexpectedly placed in the experiment section.
4. Not compare the proposed methods with that proposed in recent years. The latest baselines are from the following years, 2017, 2020, 1997.

---

> ### Author Response · Authors · 2024-12-09
> **Response to Review**
>
> Dear reviewer Dovy,
>
> Thank you for your thorough review. We are currently in the process of implementing your suggestions. Although we did not finish this work, we want to comment on your suggestions while we work on them. For the unfinished parts, we will update the paper as soon as possible and notify you about it.
>
> >W1: The major weakness is the language presentation of this paper...
>
> Thank you for pointing this out. We agree that several sentences in the paper were overly complex and difficult to read. We now revised the text and shortened many long sentences. We removed several linking words and unnecessary details to enhance the reading flow.
>
> > W2: The concept of the proposed task is poorly defined in the Introduction section...
> C2: The problem statement might need to be put in an independent section, and improve the clarity of the problem statement in the introduction...
>
>  Thank you for this suggestion. We revised the definition of the SETI task in the introduction to make it easier to understand. Moreover, we divided the introduction into multiple subsections.
>
> > W3: The organization of the sections is confusion and do not well paralleled with the introduction...
>
> We understand that the description of the TreeTransformer is extensive. However, we describe the TreeTransformer with such great detail because it is a rather uncommon model and we are afraid that our work is difficult to understand without this description. Moreover, we want the reader to understand TreePointerNet without having to read all necessary background from the original publication of the TreeTransformer first.
> With regard to the data generator, we agree that it should not be placed in the experiments section. We now moved it into its own section for better readability.
>
> >C3: The formulation expression of the "ME" operation need to be unified over the whole paper for more clearly presentation...
>
> We fixed the notation of the many-to-one embeddings and now consistently use ME throughout the paper.
>
> > W4: Not compare the proposed methods with that proposed in recent years...
> C4: Investivate more recent methods about the axiom proving problem, and represent the experiment results woth more detailed tables and figures...
>
> Thank you for pointing this out. We will add experiments with state-of-the-art LLMs on our dataset to the paper as soon as they are finished. We will also add more details to the presentation of the results as soon as we have the additional baselines.
>
> Thank you again for your feedback and we look forward to the discussion.

---

> ### Author Response · Authors · 2024-12-24
>
> Dear reviewer Dovy,
>
> Thank you again for your very thorough review. We have uploaded a new revision where we addressed your points to the best of our knowledge. In addition to the changes and answers we explained in our previous comment, we now additionally addressed the following points:
>
> > W1: The major weakness is the language presentation of this paper...
>
> We further overworked the text to make it easier to understand. In addition, we added figures to describe the data generation process (figure 3 and 4) and our many-to-one embeddings (figure 2).
>
> > W2: The concept of the proposed task is poorly defined in the Introduction section... C2: The problem statement might need to be put in an independent section, and improve the clarity of the problem statement in the introduction...
>
> We further revised the definition of the SETI task and put it into its own subsection inside the introduction for better readability.
>
> > W3: The organization of the sections is confusion and do not well paralleled with the introduction...
>
> We changed the structure of the paper to better parallel with the introduction. Precisely, we first describe our model and then the data generation, similarly to how we introduce our contributions in the introduction.
>
> > W4: Not compare the proposed methods with that proposed in recent years... C4: Investivate more recent methods about the axiom proving problem, and represent the experiment results woth more detailed tables and figures...
>
> We experimented with LLMs as additional baselines for the SETI task, but did not find them to be able to solve it. Thus, we did not include them as additional baseline to our main experiments. Please see appendix F for a description of our experiments and findings.
> We also revised the figures in our results section for better readability.
>
> Thank you again for your review and we look forward to your feedback.

---

### Review · Reviewer_EmxS · 2024-11-25

**Summary Of Contributions:**

**Summary**

The paper introduces the Stepwise Equation Transformation Identification (SETI) task, which is designed to generate a sequence of axiomatic steps to transform one mathematical expression into another equivalent form. This task addresses limitations in existing research that often treats mathematical equivalence as a binary classification without providing insight into intermediate steps.

**Contributions**

1. **TreePointerNet Model**: The paper proposes TreePointerNet, which combines hierarchical tree-structured transformers with pointer networks and many-to-one embeddings to predict transformation steps efficiently.
2. **Dataset Development**: The authors generate a new dataset comprising diverse equations, transformations, and varying complexities, ensuring robustness and benchmarking.

**Audience:**

Yes

**Broader Impact Concerns:**

No need.

**Claims And Evidence:**

Yes

**Requested Changes:**

1. Incorporate Experiments with Contemporary LLMs: add experimental baselines using large language models (LLMs) with Chain of Thought (CoT) prompting techniques.
2. Update Baseline Models: I suggest the author include more competitive models. Maybe compare the GPT4-o1 with TreePointerNet on one or two examples selected from the SETI task.
3. Discuss Broader Implications: add a section discussing the potential applications of TreePointerNet in education, research, and real-world symbolic computation tools.

**Strengths And Weaknesses:**

**Stengths**

1. The framework and SETI task can be employed in symbolic mathematics to provide interpretable intermediate steps for mathematical problem-solving.
2. The model significantly outperforms baselines (such as sequential transformers and LSTMs) on accuracy, especially for equations requiring multiple transformation steps.

**Weaknesses**

1. Lack of Integration with Contemporary LLM Approaches:
   1. While the paper introduces an innovative task and architecture, it does not explore how large language models (LLMs), such as LLaMA and Qwen could be adapted to the SETI task. These models are much more popular in recent research areas and daily lifes.
   2. Current advances in **Chain of Thought (CoT)** prompting, which allow LLMs to generate intermediate reasoning steps, could have been explored as a baseline or supplementary approach. This omission leaves a gap in comparing TreePointerNet to state-of-the-art methods for interpretability and stepwise reasoning.

---

> ### Author Response · Authors · 2024-12-09
> **Response to Review**
>
> Dear reviewer EmXS,
>
> Thank you for your thorough review. We are currently in the process of implementing your suggestions. Although we did not finish this work, we want to comment on your suggestions while we work on them. For the unfinished parts, we will update the paper as soon as possible and notify you about it.
>
> > C1 Incorporate Experiments with Contemporary LLMs... C2 Update Baseline Models...
>
> Thank you for pointing this out. One reason we did not initially compare our research to LLMs was the large discrepancy with regard to the number of parameters between our TreePointerNet and state-of-the-art LLMs. However, we will add experiments using LLMs with CoT prompting on our dataset and add the results as soon as they are finished.
>
> > C3 Discuss Broader Implications...
>
> Thank you for this suggestion. We think that this discussion is very beneficial to complete the motivation for our task. Thus, we added a new section to our paper (section 7) where we discussed possible applications and future research directions.
>
> Thank you again for your feedback and we look forward to the discussion.

---

> > ### Author Response · Authors · 2024-12-24
> >
> > Dear reviewer EmxS,
> >
> > Thank you again for your very thorough review. We have uploaded a new revision where we addressed your points to the best of our knowledge. In addition to the changes and answers we explained in our previous comment, we now additionally addressed the following points:
> >
> > >C1 Incorporate Experiments with Contemporary LLMs... C2 Update Baseline Models...
> >
> > We explored OpenAI's o1 model on a small subset from our data and found it to be able to output pairs of axioms and positions. We also found it to be able to solve equations requiring one transformation step.
> > We also experimented with open-source LLMs on the SETI task, but did not find them to be able to solve it. Thus, we did not include them as additional baselines to our main experiments. Please see appendix F for a description of our experiments and findings.
> >
> > Thank you again for your review and we look forward to your feedback.

---

### Review · Reviewer_XrD7 · 2024-11-27

**Summary Of Contributions:**

Introduction of SETI Task: The authors propose a novel task named Stepwise Equation Transformation Identification (SETI), which focuses on identifying axiomatic steps to transform one mathematical equation into an equivalent form iteratively.

TreePointerNet Architecture: They design a novel neural network, TreePointerNet, which leverages:tree-structured transformer model, a pointer generator for identifying transformation positions, and custom embeddings for distinguishing mathematical tokens.

New Dataset Creation: The authors develop and release a dataset specifically tailored for the SETI task, featuring equations of varying complexity and up to five transformation steps.

Experimental Benchmarking: Extensive comparisons against strong baselines (e.g., SeqPointer, Transformer, and LSTM models) and ablation studies demonstrate the superior performance of TreePointerNet.

**Audience:**

Yes

**Broader Impact Concerns:**

Scalability Challenges
Increasing Computational Complexity: The model’s performance decreases significantly as tree depth and transformation steps increase. While TreePointerNet is efficient relative to baselines, its hierarchical embeddings and attention mechanisms may still pose challenges when deployed in resource-constrained environments or scaled to handle millions of equations.

Interpretability and Trust Issues
Opaque Decision-Making: The model’s internal mechanisms (e.g., hierarchical embeddings and attention weights) are not fully interpretable, which might make it difficult for users to trust its predictions.

Limited Generalizability
Restricted Applicability Across Mathematical Domains: The reliance on a predefined set of axioms and tree structures limits the model’s ability to generalize to new mathematical domains or problems that involve different rules or representations.

**Claims And Evidence:**

Yes

**Requested Changes:**

Strengthen Theoretical Explanation
Discuss the theoretical advantages of TreePointerNet’s architectural components (e.g., tree structures, hierarchical embeddings, and pointer generation) over baseline models such as SeqPointer or Transformers.

Provide Comprehensive Comparisons of Efficiency
Compare TreePointerNet’s parameter count and computational complexity with baseline models. Clearly show how its efficiency is achieved without sacrificing accuracy.

Conduct Deeper Error Analysis
Investigate and categorize the root causes of misclassifications, particularly in the cases of structurally similar trees or transformations.
For instance:
Are the errors related to specific components, such as hierarchical embeddings or attention mechanisms?
How do tree depth or transformation complexity influence error rates?

Clarify and Enhance Novelty
Clearly articulate which aspects of TreePointerNet are novel compared to existing techniques (e.g., tree transformers, pointer networks).
Expand the related work section to include a more detailed comparison with similar tree-based models or pointer networks. Discuss how TreePointerNet’s contributions build upon or differ from these prior approaches.

**Strengths And Weaknesses:**

Strength:

Clear Motivation: The work tackles a well-defined problem in symbolic mathematics, emphasizing iterative, interpretable transformation steps over direct equivalence classification.

Novel Architecture: TreePointerNet effectively integrates hierarchical structural information and positional embeddings, outperforming baselines.


weaknesses:

Insufficient Mathematical or Algorithmic Explanation: The paper primarily relies on empirical results to demonstrate the effectiveness of TreePointerNet, but it lacks a systematic theoretical analysis explaining why the model outperforms baseline models (e.g., SeqPointer) on specific tasks.
For example, the paper does not clearly explain how TreePointerNet's architectural components (e.g., tree structures, hierarchical embeddings) theoretically enhance its ability to handle deep transformations.

Unclear Fairness in Complexity and Parameters: The paper claims efficiency for TreePointerNet but does not comprehensively analyze its parameter count, computational complexity, or inference speed compared to baselines. This omission makes it difficult to fairly evaluate the trade-offs.

Shallow Analysis of Misclassifications: While the paper identifies some misclassified axioms and provides examples (e.g., structurally similar trees), it does not deeply investigate the root causes of these errors.
For instance: Are the errors due to failures in hierarchical embeddings or attention mechanisms? How do misclassifications differ across trees of varying depth and complexity?

Incremental Contribution: The main innovations of TreePointerNet (e.g., tree-structured transformers, pointer generation, hierarchical embeddings) are primarily adaptations or combinations of existing techniques, rather than entirely novel contributions.

---

> ### Author Response · Authors · 2024-12-09
> **Response to Review**
>
> Dear reviewer XrD7,
>
> Thank you for your thorough review. We are currently in the process of implementing your suggestions. Although we did not finish this work, we want to comment on your suggestions while we work on them. For the unfinished parts, we will update the paper as soon as possible and notify you about it.
>
> >W1: Insufficient Mathematical or Algorithmic Explanation... C1: Strengthen Theoretical Explanation Discuss the theoretical advantages of TreePointerNet’s architectural components...
>
> Thank you for pointing this out. So far, we shortly discussed the motivation for our many-to-one embeddings in section 2.3 and the motivation and necessity for the pointer component in section 2.2 and 5.2. However, we see that our current explanations are very brief. We will add a new section where we discuss the theoretical motivation behind our components in larger detail.
>
> >W2: Unclear Fairness in Complexity and Parameters... C2: Provide Comprehensive Comparisons of Efficiency
>
> Our results indicate that TreePointerNet (TPN) achieves better results than a transformer or SeqPointer while requiring less parameters. This is the result of our hyperparameter study where we used the same search space for all transformer based models (including TPN). We report the number of parameters in section 4.5. To better quantify the influence of the number of parameters we will evaluate a transformer with the same hyperparameters we used for TPN and hence the same number of parameters as TreePointerNet and add this result.
> With regard to the computational complexity, there is no difference between a transformer and TreePointerNet. It is $\mathcal{O}(n^2)$ for both models. We added a detailed deduction of the complexity of TreePointerNet in our revised paper in section A.2.
>
> >W3: Shallow Analysis of Misclassifications... C3: Conduct Deeper Error Analysis
>
> We are unsure if we understand your question correctly. We analyzed the influence of the tree depth and transformation complexity in section 5.4 and present the results in table 1. We can have a look at the individual components of TPN to see if we find a reason for its above-average failure on some axioms, yet there is no guarantee that an interpretable explanation exists. It’s inherently hard to clearly identify why a transformer-based model makes a certain decision because many attention heads are computed. Moreover, even if a certain pattern can be observed in the activations of a trained model, there is no guarantee that this pattern reemerges in another trained model with the same architecture.
>
> >W4: Incremental Contribution... C4: Clarify and Enhance Novelty...
>
> We revised our related work section and added a discussion with regard to the similarity of other tree-based models and pointer networks with our TreePointerNet.
>
> Thank you again for your feedback and we look forward to the discussion.

---

> > ### Author Response · Authors · 2024-12-24
> >
> > Dear reviewer XrD7,
> >
> > Thank you again for your very thorough review. We have uploaded a new revision where we addressed your points to the best of our knowledge. In addition to the changes and answers we explained in our previous comment, we now additionally addressed the following points:
> >
> > > W1: Insufficient Mathematical or Algorithmic Explanation... C1: Strengthen Theoretical Explanation Discuss the theoretical advantages of TreePointerNet’s architectural components...
> >
> > We now overworked our section 2 and added more detailed motivations for the introduction of the architectural components of TreePointerNet.
> >
> > >W2: Unclear Fairness in Complexity and Parameters... C2: Provide Comprehensive Comparisons of Efficiency
> >
> > Beside the discussion of the complexity in appendix A.2, we now compare its parameter efficiency in appendix A.3. In short, we show that inputting the data as trees is beneficial for reducing the number of parameters. To do so, we compare a TreeTransformer and a transformer having the same number of trainable parameters and find the TreeTransformer to outperform the transformer on sequential input.
> >
> > Thank you again for your review and we look forward to your feedback.

---

### Review · Reviewer_nGRS · 2024-11-30

**Summary Of Contributions:**

**1. Introduction of the SETI Task:**

The paper introduces a novel task called Stepwise Equation Transformation Identification (SETI), which focuses on iteratively transforming a mathematical expression into an equivalent form using a sequence of axiomatic steps.  This task is significant, as it bridges the gap between equivalence classification and interpretable reasoning, enabling deeper exploration into step-by-step mathematical problem-solving.

**2. TreePointerNet Architecture:**

The authors propose TreePointerNet, a novel neural network architecture that effectively combines a TreeTransformer-based encoder, a pointer generator network for position identification, and a Many-To-One Embedding Layer.  This integration enables efficient and accurate modeling of hierarchical mathematical transformations.

**3.  Dataset Creation:**

A comprehensive dataset tailored for the SETI task is presented, featuring equations with varying complexities and up to five transformation steps.  The dataset, built on previous generation techniques (e.g., Wankerl et al., 2023), is unique in its explicit focus on iterative transformations, filling a gap in symbolic mathematics datasets.

**4. Experimental Benchmarks:**

Extensive experiments demonstrate the superior performance of TreePointerNet over multiple baselines, with robust generalization to deeper parse trees and higher transformation complexity.

**Audience:**

Yes

**Broader Impact Concerns:**

**1. Scalability Challenges for Complex Problems:**

The model’s performance degradation with increasing transformation steps is evident. In competitive mathematical problem-solving, proofs often require ten or more steps. Whether this lightweight tree-based model can handle such problems remains uncertain and warrants further exploration.

**Claims And Evidence:**

Yes

**Requested Changes:**

**1. Reorganize the Paper for Clarity:**

Improve the overall structure of the paper.  For example, integrate figures to explain the network structure concisely, focusing on the pointer mechanism and the Many-To-One Embedding Layer (ME) for better readability.

**2. Simplify Dataset Generation Explanation:**

Use figures to illustrate the dataset construction process, reducing the reliance on textual explanations.

**3. Enhance Motivation and Applications of the SETI Task:**

Provide a more thorough discussion on the motivation behind introducing SETI and highlight its real-world applications.

**4. Standardize Terminology:**

Ensure consistent usage of key terms like “ME” throughout the paper.  For instance, “me” is used in lowercase on page 6 but appears as “ME” in uppercase in Figure 3.  Consider using italicization or another consistent format to denote this component.

**Strengths And Weaknesses:**

**Strengths:**

**1. Novel Task Design:**

SETI is a task that shifts the focus from simple equivalence classification to iterative, interpretable reasoning steps. This direction aligns with real-world applications where stepwise explainability is essential, such as in automated theorem proving.

**2. Innovative Architecture:**

TreePointerNet demonstrates strong architectural novelty through its integration of tree-structured input, pointer mechanisms, and Many-To-One embeddings. These components are shown to enhance performance and robustness.

**3. Scalable Dataset Construction:**

The dataset construction process is highly efficient and automated, leveraging extensions of existing generation frameworks.    The diversity of included transformations (polynomials, trigonometry, logarithms, etc.) ensures broad applicability.

**4. Lightweight Model with Robust Performance:**

TreePointerNet requires significantly fewer parameters than baselines while achieving state-of-the-art performance.    Its robustness to increasing transformation steps and tree depth is particularly noteworthy.

**Weaknesses:**

**1. Unclear Motivation and Application Scenarios for the SETI Task:**

The paper lacks a detailed discussion on the motivation for introducing the SETI task and its potential real-world applications. This requires further elaboration.

**2. Limited Visualization:**

Despite the technical depth of the paper, many sections rely heavily on textual descriptions. For example, the explanation of data generation and the TreePointerNet architecture could be significantly enhanced with additional figures or flowcharts.

**3. Outdated Baselines:**

While TreePointerNet is based on tree neural networks, the baselines used for comparison rely on traditional neural networks, some of which are outdated.  This raises concerns about the fairness of the comparisons.

---

> ### Author Response · Authors · 2024-12-09
> **Response to Review**
>
> Dear reviewer nGRS,
>
> Thank you for your thorough review. We are currently in the process of implementing your suggestions. Although we did not finish this work, we want to comment on your suggestions while we work on them. For the unfinished parts, we will update the paper as soon as possible and notify you about it.
>
> >W1: Unclear Motivation and Application Scenarios for the SETI Task... C3: Enhance Motivation and Applications of the SETI Task...
>
> Thank you for this suggestion. We think that a discussion of potential use cases is very beneficial to complete the motivation for our task. Thus, we added a new section to our paper (section 7) where we discussed possible applications and future research directions. Among others, we think that our model could be beneficial in educational settings for selecting individually fitting exercises to students or providing hints in an interactive teaching tool.
>
> >W2: Limited Visualization... C1: Reorganize the Paper for Clarity... C2: Simplify Dataset Generation Explanation...
>
> Thank you for pointing this out. We agree that illustrations can be very helpful to understand the data generation and model. We will add illustrations to our paper describing the data generation and details of our architecture. We also improved the structure of the paper by moving the dataset generation into its own section and dividing the introduction into subsection.
>
> >W3: Outdated Baselines...
>
> Thank you for pointing this out. We will add experiments with state-of-the-art LLMs (Qwen/LlaMa) on our dataset to the paper as soon as they are finished.
>
> >C4: Standardize Terminology
>
> We fixed the notation of the many-to-one embeddings and now consistently use ME throughout the paper.
>
> Thank you again for your feedback and we look forward to the discussion.

---

> ### Author Response · Authors · 2024-12-24
>
> Dear reviewer nGRS,
>
> Thank you again for your very thorough review. We have uploaded a new revision where we addressed your points to the best of our knowledge. In addition to the changes and answers we explained in our previous comment, we now additionally addressed the following points:
>
> >W2: Limited Visualization... C1: Reorganize the Paper for Clarity... C2: Simplify Dataset Generation Explanation...
>
> We now added visualizations of the data generation process (figure 3 and 4) and our many-to-one embeddings (figure 2) to the paper. The pointer component is depicted in figure 9. We further revised the description of our model, the data generation and the SETI task in sections 1-3.
>
> >W3: Outdated Baselines...
>
> We experimented with LLMs on the SETI task, but did not find them to be able to solve it. Thus, we did not include them as additional baseline to our main experiments. Please see appendix F for a description of our experiments and findings.
>
> Thank you again for your review and we look forward to your feedback.

---

### Review · Reviewer_Z11v · 2024-12-02

**Summary Of Contributions:**

1. Introduced a **new task**, namely Stepwise Equation Transformation Identification (SETI): given two equivalent math expressions, generate a sequence of (axiom, position) pairs to transform one expression into the other;
2. Constructed a **dataset** for model training and evaluation on SETI;
3. Implemented a **novel model**, TreePointerNet, which is specifically designed for SETI by incorporating a tree-structured transformer with a copy-pointer mechanism and a custom embedding. Ablation experiments demonstrate the effectiveness of these enhancements.

**Audience:**

Yes

**Broader Impact Concerns:**

None.

**Claims And Evidence:**

Yes

**Requested Changes:**

1. It is advisable to refine the writing to improve its clarity and conciseness. (critical)
2. It is recommended to incorporate a literature review of symbolic methods for equivalence determination. (recommended)
3. Expanding the domains and reconsidering the axioms could enhance the generalizability and significance of this work. (critical)
4. The addition of symbolic baselines and a fairer tokenization strategy would make the experimental results more convincing. (critical)

**Strengths And Weaknesses:**

### Strengths
- Unlike previous studies, which mainly focused on determining the equivalence of math expressions, the proposed task, SETI, targets finding a sequence of symbolically verifiable steps for equivalence determination. This makes the results more reliable and interpretable.
- A domain-specific model, TreePointerNet, has been implemented for SETI. Experiment results demonstrate its effectiveness over baselines and the benefits of the ablative improvements.
- The experiments are solid. They report not only average values but also the standard deviation over 5 runs.

### Weaknesses
1. Poor organization and writing quality.
	1. The content structure is illogical. For instance, the dataset generation is placed in Sec.3. Experiment.
	2. The writing is overly verbose.
	3. There are several typos:
		1. In Sec.3.1, since $\mathbb R^+ = \\{x\in\mathbb R | x > 0\\}$, it is unnecessary to write $x, y, z \in \mathbb R^+\backslash{}\\{0\\}$
		2. In Sec.5 L1: "_To the best **o f** our knowledge, there **exits** no closely related work_", where "o f" and "exits" appear to be misspellings.
2. Absence of important related works.
	- The authors should review symbolic methods for determining the equivalence of math equations. For example, [1] can solve equations (and subsequently determine the equivalence of math expressions) in commutative (semi)rings.
3. All constants are limited to $\\{-4, -3, \dots, 3, 4\\} \cup \\{e, \pi\\}$, and variables are assumed to be $x,y,z\in \mathbb R^+\backslash{}\\{0\\}$, which may lead to
	1. Limited applicability. Such a tight limitation makes this dataset deviate far from real-world applications.
	2. The operations do not have closure under domain $\mathbb R^+$. Given constants in $\\{-4, \dots, 4\\} \cup \\{e, \pi\\}$ or $\mathbb R^+$, expressions can sometimes result in negative real numbers and zero, e.g. $\sin(4) \approx -0.76 < 0 \not \in \mathbb R^+$, which fall outside the domain of variables $\mathbb R^+$.
	3. Some "axioms" are not valid in certain cases, e.g. $\ln(1) = 0$, thus $\ln(1)^{\ln(1)} = 0^0$ is an undefined expression ($\alpha^0\leftrightarrow 1$ doesn't hold); furthermore, $\frac{\ln(1)}{1} = \frac{0}{1} = 0$, but $(\frac{1}{\ln(1)})^{-1} = (\frac{1}{0})^{-1}$ is undefined ($\alpha/\beta\leftrightarrow (\beta/\alpha)^{-1}$ doesn't hold).
4. Lack of supporting literature.
	- In the abstract, the authors state "*this problem is expensive to solve with rule-based systems due to the large search space*". In the introduction, they claim "_In practice, however, given a large enough equation and set of axioms, this quickly becomes computationally infeasible_". However, throughout the paper, the reviewer fails to find any supporting literature for these claims.
5. Insufficient discussion with Formal Theorem Proving.
	- In the introduction, the authors mention "_in remotely related settings like step-by-step solutions to word problems, only the final answer is checked, but not the predicted intermediate steps_". But in formal theorem proving, all steps are formal-verifiable.
6. Incomplete Baselines.
	- Symbolic methods should have been included.
7. The tokenization strategy for baselines may result in unfair comparisons.
	- Each token (operator, variable, and constant) is tokenized differently based on its position, as shown in Sec.3.2 "_the equation `ln(1)/1 = 0` is tokenized to `[=, /_0, ln_0, 1_0, 1_1, 0]` for the sequential baselines_". The more frequently a particular token appears, the lower the probability of generating the corresponding equation. For example, $1_0 +_0 1_1 +_1 1_2 +_2 1_3 = 4_0$ has a low generation probability. Hence, tokens that occur multiple times in one expression, such as the $+_2$ and $1_3$ in the above example, are scarce in the training set, and baselines may underfit them. This unfairness is more severe in Sec.4.3.Robustness Study.

[1] Grégoire, Benjamin, and Assia Mahboubi. "Proving equalities in a commutative ring done right in Coq." _International Conference on Theorem Proving in Higher Order Logics_. Berlin, Heidelberg: Springer Berlin Heidelberg, 2005.

---

> ### Author Response · Authors · 2024-12-09
> **Response to Review**
>
> Dear reviewer Z11v,
>
> Thank you for your very thorough review. We are currently in the process of implementing your suggestions. Although we did not finish this work, we want to comment on your suggestions while we work on them. For the unfinished parts, we will update the paper as soon as possible and notify you about it.
>
> >W1: Poor organization and writing quality... W1: It is advisable to refine the writing to improve its clarity and conciseness...
>
> Thank you for pointing this out. We agree that the data generator should not be described in the experiments section. We now moved it into its own section for better readability. We fixed the typos you mentioned. We also agree that the paper is currently too text-heavy. We will add figures to describe the data generation process and the model better.
>
> >W2: Absence of important related works... C2: It is recommended to incorporate a literature review of symbolic methods for equivalence determination...
>
> Thank you for this comment. Unfortunately, we are unsure if we understand it correctly. Determining the equivalence of mathematical expressions is only a binary classification task. In our paper, we want to find a sequence of axioms and their exact position of application to show how an expression can be transformed into another given equivalent form. Moreover, in our setting, the equivalence between these expressions is assumed and not further checked. Hence, we unfortunately do not understand why literature on rule-based systems which can determine the equivalence is important related work for our task. Could you please shed more light on this?
>
> > W3+C3: W.r.t. the limited applicability + the missing closure under $\mathbb{R}^+$
>
>  We based our work on a generator known from literature (Wankerl et al. 2023). To the best of our knowledge, this is the generator that poses the least constraints on the data properties and is best to adapt for our task. Of course, we agree that the data is not without restrictions in the sense that it covers a subset of mathematical rules and value ranges. Also, more unconstrained datasets would surely also pose an asset to the research field.
>
> Still, we argue that for our cause, the current state-of-the-art data generation approach suffices for demonstrating our model’s ability to solve the introduced SETI task in a thorough evaluation. In particular, we think that data constraints in general do not really impose restrictions on our proposed model, since it can always be retrained on more unconstrained data corpora as they become available. In short, due to our focus on task and model design, we leave the design of further improved data generators to future work.
>
> > W3 + C3: Wrt. to the undefined axioms
>
> We understand your confusion here, because the description of our generator was too brief. In fact, the generator evaluates every generated expression using the Sympy library. Here, the values of the variables are assumed to be positive real numbers. Every expression that is evaluated to Nan or infinity gets discarded and will not be present in our dataset. We initially omitted this detail for brevity, but now added it to the data generator description.
>
> > W4: Lack of supporting literature...
>
> Thank you for pointing this out. We removed these claims as they were actually out of scope for our work which does not focus on an efficiency comparison between deep learning and rule-based approaches.
>
> > W5: Insufficient discussion with Formal Theorem Proving...
>
> Thank you for pointing this out. Unfortunately, our wording was misleading because we only wanted to refer to deep-learning-based approaches to word problems. We now specify it more precisely in the paper.
>
> >W6: Incomplete Baselines...
>
> Unfortunately, this suggestion is similarly confusing to us as W2. We are aware that rule-based systems can be used to determine the equivalence of mathematical expressions and previous research (Wankerl et al., 2023) has already compared rule-based and deep-learning-based systems on this task. To the best of our knowledge, rule-based systems like Sympy, Maxima, or Coq only output if they are able to verify the equivalence or not. We do not see how they could be used to output sequences of applied axioms and their positions in a similarly fine-grained way as we do in our research. However, this would be necessary to have a fair comparison to our neural networks. Could you maybe shed some more light on this issue?

---

> > ### Author Response · Authors · 2024-12-09
> > **Response to Review (continued)**
> >
> > >W7: The tokenization strategy for baselines may result in unfair comparisons...
> >
> > This is an interesting idea. We will represent the index as a separate token (e.g., x 0, y 3) as an alternative strategy although this implies that the length of the input sequence is doubled. Moreover, the model has to learn additional relations between the input tokens.
> >
> > Yet, we want to point out that the tokenization we do for the baselines is exactly the same we use for our TreePointerNet, as can be seen in figure 1. For TreePointerNet, each leaf and non-terminal node also holds both the mathematical token as well as its index, e.g. x_4, +_0. Basically, our current tokenization strategy is a trade-off between number of input tokens and frequency of each token.
> >
> > > C4: The addition of symbolic baselines and a fairer tokenization strategy would make the experimental results more convincing.
> >
> > Thank you for this suggestion. We hope we can clarify it with W6 and W7 above.
> >
> > Thank you again for your feedback and we look forward to the discussion.

---

> > > ### Comment · Reviewer_Z11v · 2024-12-10
> > >
> > > Thank you, authors, for your detailed and prompt response. Most of my concerns have been satisfactorily addressed. However, there are still a few remaining points that require further clarification:
> > >
> > > > Response to W2: rule-based systems which can determine the equivalence
> > >
> > > > W6: Incomplete Baselines
> > >
> > > My apologies for any lack of clarity in the initial review. When I mentioned "symbolic methods," I was referring to certain techniques that could potentially be relevant to the SETI task. For instance:
> > > 1. Automatic theorem provers and SMT solvers like Z3. Z3 can determine the equivalence between two expressions and generate proof objects[3] for additional analysis. However, when dealing with trigonometric functions, exponentials, and logarithms, some additional formalization work might be necessary[1].
> > > 2. Proof Automation in formal proof assistants, such as `aesop?`[2] in Lean and `sledgehammer` in Isabelle. These can be used to search for a sequence of proof steps given a statement, in our case, the equivalence between two expressions. For example, `aesop` can generate a proof of $((2 \cdot x) \cdot x) \cdot y = (x + x) \cdot (x \cdot y)$ using `mul_assoc` (multiplication associativity) and `two_mul` ($2 \cdot n = n + n$), as demonstrated in the following Lean code:
> > > ```lean
> > > attribute [aesop safe] mul_assoc
> > > attribute [aesop norm] two_mul
> > >
> > > example (x y : Real) : ((2 * x) * x) * y = (x + x) * (x * y) := by
> > >   aesop?
> > >
> > > -- Generated proof:
> > > -- simp_all only [two_mul]
> > > -- apply mul_assoc
> > > ```
> > > Nevertheless, considering your revision in response to W4, it seems that including such a symbolic baseline might not be essential. As long as W7 is addressed, I will change the "Claims And Evidence" to "Yes". However, incorporating it could lead to a more comprehensive and in-depth comparison.
> > >
> > > > W7. Unfair comparison resulting from tokenization strategy.
> > >
> > > I appreciate your active efforts in improving the paper. Thank you for your efforts during the rebuttal process. I look forward to seeing the updated version.
> > >
> > > [1] https://github.com/Z3Prover/z3/issues/680
> > >
> > > [2] Limperg, Jannis, and Asta Halkjær From. "Aesop: White-box best-first proof search for Lean." _Proceedings of the 12th ACM SIGPLAN International Conference on Certified Programs and Proofs_. 2023.
> > >
> > > [3] Böhme, Sascha. "Proof reconstruction for Z3 in Isabelle/HOL." _7th International Workshop on Satisfiability Modulo Theories (SMT’09)_. 2009.

---

> ### Author Response · Authors · 2024-12-24
>
> Dear reviewer Z11v,
>
> Thank you again for your very thorough review. We have uploaded a new revision where we addressed your points to the best of our knowledge. In addition to the changes and answers we explained in our previous comment, we now additionally addressed the following point:
>
> > W1: Poor organization and writing quality... W1: It is advisable to refine the writing to improve its clarity and conciseness...
>
> We revised our writing and shortened many long sentences. We restructured the introduction, moved the data generation into its own section and added figures to visualize the data generation (figure 3 and 4) and the many-to-one embeddings (figure 2).
>
> > W7. Unfair comparison resulting from tokenization strategy.
>
> We now evaluated a different tokenization strategy where the index is represented independent from the token. However, since it did not prove to be superior we leave the tokenization strategy of our main experiments unchanged. Please find the details about this strategy in section 4.1 and the results of our experiments on the new strategy in appendix E of our new revision.
>
> Thank you again for your review and we look forward to your feedback.

---

> > ### Comment · Reviewer_Z11v · 2024-12-25
> >
> > Thank you for the detailed response and experiments. I'll update the "Claims And Evidence" to "Yes".
> >
> > BTW, in Table 13, within the "_Number of Transformation Steps / 1_" column, it appears that there might be a typo. SeqPointer attains a performance of 88.58% but TreePointerNet (88.43%) is marked in bold. Could you please double-check this?

---

### Decision · Action_Editor_6c5D · 2024-12-29

**Recommendation:** Accept as is

**Comment:**

Please see above.

After the rebuttal, the reviewers are positive about the manuscript and recommend acceptance of the paper. The authors are recommended to further consider enhancing the versatility of the task and using more up-to-date baselines.

**Audience:**

Yes.

**Claims And Evidence:**

Yes.

This paper Introduced a new task, namely Stepwise Equation Transformation Identification (SETI): given two equivalent math expressions, generate a sequence of (axiom, position) pairs to transform one expression into the other. It also constructed a dataset for model training and evaluation on SETI. The authors also implemented a novel model, TreePointerNet, which is specifically designed for SETI by incorporating a tree-structured transformer with a copy-pointer mechanism and a custom embedding. Ablation experiments demonstrate the effectiveness of these enhancements.